# Decisive role of water and protein dynamics in residence time of p38α MAP kinase inhibitors

Tatu Pantsar[1,2], Philipp D. Kaiser[3], Mark Kudolo[1], Michael Forster[1], Ulrich Rothbauer [3,4,5] &
Stefan A. Laufer [1,5,6✉]

Target residence time plays a crucial role in the pharmacological activity of small molecule inhibitors. Little is known, however, about the underlying causes of inhibitor residence time at the molecular level, which complicates drug optimization processes. Here, we employ all-atom molecular dynamics simulations (~400 μs in total) to gain insight into the binding modes of two structurally similar p38α MAPK inhibitors (type I and type I½) with short and long residence times that otherwise show nearly identical inhibitory activities in the low nanomolar $IC_{50}$ range. Our results highlight the importance of protein conformational stability and solvent exposure, buried surface area of the ligand and binding site resolvation energy for residence time. These findings are further confirmed by simulations with a structurally diverse short residence time inhibitor SB203580. In summary, our data provide guidance in compound design when aiming for inhibitors with improved target residence time.

[1] Department of Pharmaceutical and Medicinal Chemistry, Institute of Pharmaceutical Sciences, Eberhard Karls Universität Tübingen, Auf der Morgenstelle 8, 72076 Tuebingen, Germany. [2] School of Pharmacy, Faculty of Health Sciences, University of Eastern Finland, Yliopistonranta 1, 70210 Kuopio, Finland. [3] NMI Natural and Medical Sciences Institute at the University of Tuebingen, Markwiesenstrasse 55, 72770 Reutlingen, Germany. [4] Pharmaceutical Biotechnology, Eberhard Karls University Tuebingen, Markwiesenstrasse 55, 72770 Reutlingen, Germany. [5] Cluster of Excellence iFIT (EXC 2180) "Image-Guided and Functionally Instructed Tumor Therapies", University of Tuebingen, 72076 Tuebingen, Germany. [6] Tuebingen Center for Academic Drug Discovery & Development (TüCAD2), 72076 Tuebingen, Germany. ✉email: stefan.laufer@uni-tuebingen.de

Lack of efficacy is one of the major hurdles in clinical drug development resulting in low success rates in clinical trials[1–4]. Efficacy can be further divided in pharmacodynamic and disease efficacy[5]. One approach to tackle the pharmacodynamic efficacy problem is the target residence time ($\tau = 1/k_{off}$) concept, which, when increased, can also improve the actual efficacy in vivo[6–8]. Therefore, addressing the target residence time could provide an important solution to overcome the lack of efficacy of small molecules at an early stage in drug development. However, it should not be considered as the only parameter in the drug optimization or at the expense of other important aspects such as pharmacokinetics[9].

Earlier computational work investigating residence times has been mainly focusing on association/dissociation pathways of small molecules[10–15]. These studies applied molecular dynamics (MD) simulations with enhanced sampling methods to enable relevant timescales required for the association/dissociation process[16–19]. Another approach is based on a combination of MD simulations with Brownian dynamics and milestoning methodology[20,21]. While there has been much recent focus on predicting the absolute residence time of a small molecule, less attention has been paid to the actual protein–ligand complex and its role in relation to residence time. Although it would be useful to predict the residence time of a ligand, these approaches currently fail to provide guidance to the exact compound design process (how to modify the compound). A deeper understanding of the atomistic level processes associated with shorter and longer residence times in protein–ligand complexes could provide better insights here to facilitate the process of ligand design.

Among the most widely pursued drug target class, the protein kinases[3], p38α MAPK represents an intensively studied kinase, for which numerous inhibitors have been developed to date, but still no approved drug exists[22]. Conformational dynamics of this kinase on a longer timescale have been studied in the context of its activation process by metadynamics[23] and in classical simulations focussing mainly on apo p38α MAPK[24]. Additionally, p38α MAPK inhibitor dissociation has been investigated by simulations. For instance, dissociation pathways of type I and type II inhibitors were evaluated utilizing accelerated MD[25] and unbinding of a type II inhibitor was studied by the metadynamics approach[11]. Moreover, dissociation of selected type I and type II inhibitors were studied by steered MD[10]. While this approach provided useful prediction of the residence time on a larger scale (hours vs. minutes), it was still not possible to obtain more accurate information on the exact residence times.

In this context, we have previously reported that among p38α MAPK inhibitors, the type I½ inhibitor regulatory-spine (R-spine) interaction is important for increased target residence time[26–28]. However, these type I½ inhibitors have not been studied by the means of long timescale simulations to date and a precise understanding of the underlying mechanisms is still incomplete.

Here we applied unbiased classical long timescale MD simulations to investigate behavior of the protein–ligand complex of p38α MAPK and two inhibitors, representing a first and a second generation dibenzosuberone-based inhibitor (type I and type I½) originated from Skepinone-L[29]. These compounds provide an excellent opportunity for detailed study, as they differ significantly in their residence time but are structurally similar and show comparable target inhibition, as demonstrated in biochemical assays with isolated enzymes. Our aim was to obtain deeper insights into the potential differences in protein–ligand complex behavior of these structurally related inhibitors at the atomistic level. We anticipated that a better understanding of this process can provide general principles, which could be applied in the compound design process when aiming for improved target residence time. Furthermore, we complemented and confirmed our key findings with simulations of a structurally diverse fast residence time inhibitor **SB203580**, well-tempered metadynamics simulations, as well as with swapped-ligand simulations. In this study, we identified protein conformational stability and water as decisive elements that demonstrated considerable differences among compounds with short and long residence time.

## Results

**Structurally similar inhibitors with different residence times.** Compounds **1** and **2** used in this study are structurally identical, except **2** has an additional R-spine interacting moiety of a type I½ inhibitor (Fig. 1a). Although these compounds display virtually identical $IC_{50}$ values in isolated enzyme inhibition assays, we noticed that **2** shows a ~3.5 to ~4.4-fold higher potency to inhibit p38α MAPK in a cell-based MK2 translocation assay (Fig. 1b and Supplementary Figs. 1 and 2) using two different human cell models. We have earlier determined the target residence time for these compounds by a surface plasmon resonance (SPR) assay with the active kinase[28], where a remarkable two orders of magnitude longer residence time was observed for **2**. Here, we re-evaluated the residence time of these compounds in an orthogonal assay, utilizing a fluorescence polarization (FP) based approach[30] Although the absolute difference of the residences time is decreased, we still observed a substantial (~11-fold) difference that is more consistent with the differences in efficacy observed in the cell-based MK2 translocation assay. It is conceivable that the FP assay better reflects the actual biological situation because it contains ATP and substrate that were not present in the SPR assay.

The discrepancy of these two compounds in their residence times and biological activities motivated us to reveal in more detail their potential differences at atomistic level in terms of their binding modes in complex with p38α MAPK. To this end, we conducted MD simulations that resulted in total of 91 μs simulation data for **1** and 86 μs for **2**. Both compounds display stable binding throughout all simulations (Supplementary Fig. 3); therefore, these simulation data describe solely the stable binding mode and behavior of the protein–ligand complex.

The compounds' shared structural scaffold appears in almost identical position when bound to the kinase, with only a slight shift in the position of their difluorophenyl ring (Fig. 1c and Supplementary Fig. S4). This scaffold is anchored to p38α MAPK by a stable hinge interaction from the dibenzosuberone (H-bond to Met109) (Fig. 1d). The NH which connects dibenzosuberone and difluorophenyl rings displays water-bridged interactions to Asp168 and to the backbone of Phe169 with **1**, whereas a more frequent water bridge appears only to Asp168 with **2**. The catalytic lysine of p38α MAPK, Lys53, displays a π–cation interaction with the difluorophenyl ring. This π–cation interaction is more frequent with **2** (51%) compared to **1** (27%). In the solvent interface, the proximal 2-morpholinoethylamide, which is fluctuating throughout the simulations, appears somewhat more stable with **1** (more conserved H-bond interactions). Of note, in the crystal structure the electron density is disordered for this moiety (Supplementary Fig. 5), which is in agreement with the observed simulation fluctuations. The additional type I½ residue of **2**, thiophene-2-carboxamide, displays a stable H-bond from its carbonyl-oxygen to the backbone of Asp168 (99%). Moreover, the NH displays a H-bond to Glu71 of the αC-helix and a stable π–π stacking interaction between the thiophene and Phe169 exists (86%).

One key difference observed in the interactions among the compounds with their shared scaffold is related to their different binding modes. With **1**, which is bound in DFG-out/inactive

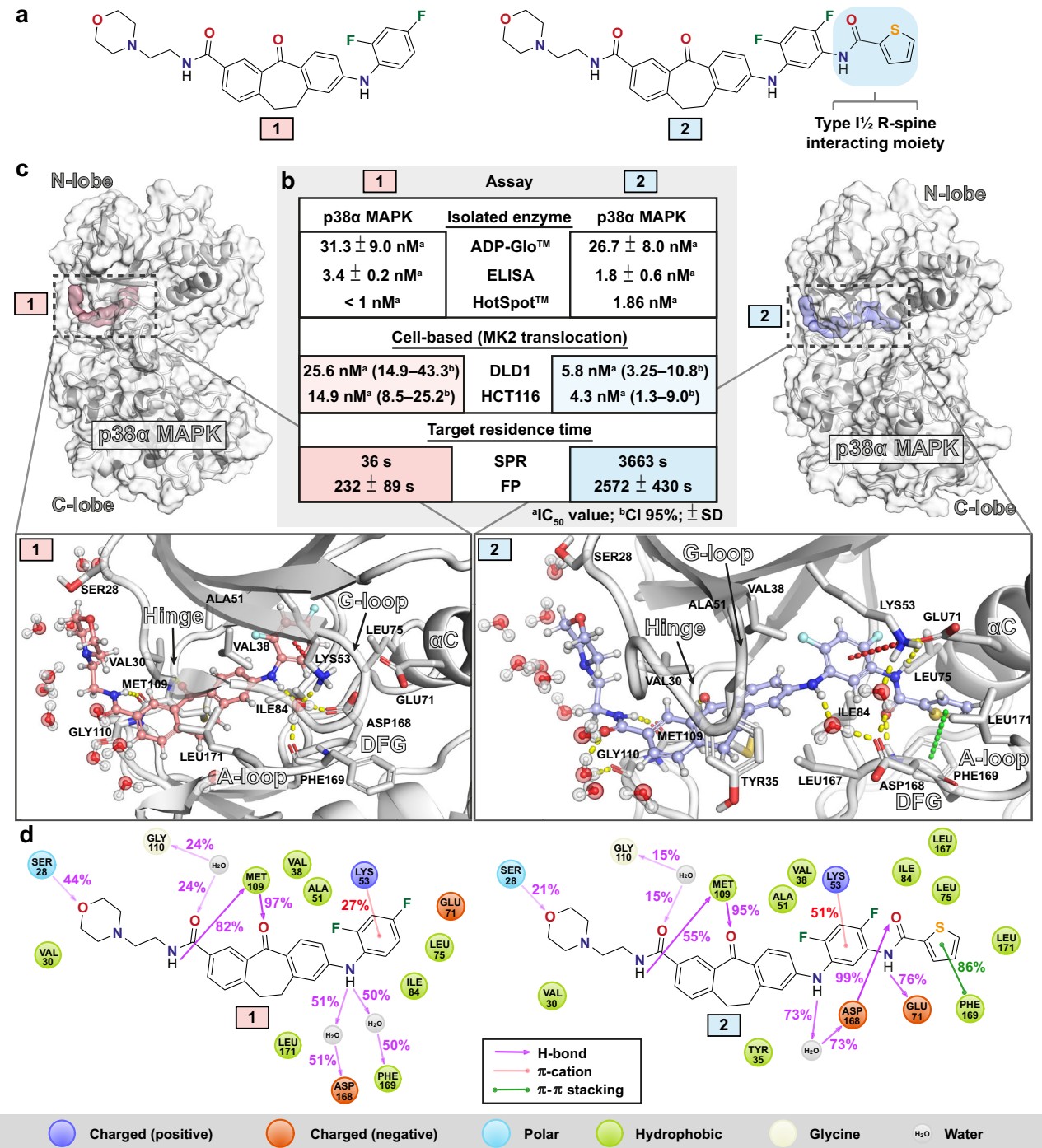

**Fig. 1 Compounds 1 and 2 biological activities and binding modes to p38α MAPK. a 1** and **2** are structurally identical, except **2** has an additional thiophene-2-carboxamide group. **b** Biological activity of **1** and **2** in selected assays. Data for ELISA and SPR were reported earlier by our group[27,28]. (ADP-Glo, ELISA, DLD1, HCT116, FP: n = 3; SPR n = 2; HotSpot, n = 1). **c** Binding modes of **1** and **2** shown with representative snapshots obtained from MD simulations (the snapshots were selected to reflect the observed protein ligand interactions; see **d**). In the figure that is illustrating the whole kinase, molecular surface of p38α MAPK is displayed in white, and molecular surface for **1** and **2** are shown in light red and light blue, respectively. In the close-up image of the binding site, the key interaction residues are shown with sticks together with solvent molecules in the close proximity of the ligand. Ligands are displayed with ball and stick representations using light red and light blue colors for carbon-atoms of **1** and **2**, respectively. H-bonds shown with yellow, π–cation with red and π–π stacking with green dashed lines. **d** 2D-representation of protein–ligand interaction frequencies. Displayed are the most frequent H-bond, π–cation, and π–π stacking interaction frequencies (interactions with >15% frequency are shown). Source data are provided as a Source Data file.

A-loop configuration, shielding of the hydrophobic dibenzosuberone scaffold from the solvent is conducted by Leu171 from the A-loop (Fig. 1c and Supplementary Fig. 4). Interestingly, this hydrophobic shielding is replaced by Tyr35 from the G-loop with

**2** (Fig. 1c and Supplementary Fig. 6). In this case shielding by Leu171 is not possible as it exists in DFG-in/active A-loop configuration, where the Leu171 is participating in forming a lipophilic pocket to accommodate the hydrophobic thiophene

group (Fig. 1c and Supplementary Fig. 4). In the crystal structure of p38α MAPK in complex with **2** (PDB ID: 5tbe) disordered residues are present in the G-loop (Supplementary Fig. 3A), which suggests instability in the observed crystal conformation of this region. Moreover, in a recently published double phosphory-lated p38α MAPK–Skepinone-L–ATF2 structure (PDB ID: 6zqs), Tyr35 exists in similar configuration as seen in simulations (Supplementary Fig. 5B)[31]. This provides further rationale for the considerable conformational rearrangement of the G-loop observed in the simulations of **2**.

**Conformational dynamics of p38α MAPK is stabilized by 2.** We next shifted our attention to protein conformational dynamics in the p38α MAPK-inhibitor complexes. To disclose relevant conformations, we conducted a Markov State Modeling (MSM) approach. This methodology is able to capture relevant long timescale kinetic conformational states, the metastable states, of the protein–inhibitor complex[32–34].

For **1**, MSM identified three metastable states: **1-$S_{1-3}$** (Fig. 2). All three states appear considerably populated. Even for the least populated state, **1-$S_1$**, the equilibrium probability ($\pi_1$) is above 0.25, which means that the system populates this metastable state more than a quarter of the time (25%). The probability for the second most populated state **1-$S_2$** is around one third (32%), whereas for the most dominant state **1-$S_3$** almost 43%. The three metastable states can be easily distinguished by the configuration of the A-loop (Fig. 2c). These conformational shifts related to A-loop are clearly demonstrated by spatial orientation of the Thr175 of the A-loop. While the average distance difference between the starting conformation of the simulation and the representative structures for the Cα-atom of Thr175 is only 1.8 Å for **1-$S_3$**, it is 7.2 Å and 9.3 Å for **1-$S_1$** and **1-$S_2$**, respectively. The conformational difference between these two systems is also evident from the average distance of 3.5 Å between the Thr175 Cα-atoms of **1-$S_1$** and **1-$S_2$** (Supplementary Fig. 7). Altogether, these observations highlight that the A-loop of p38α MAPK is constantly fluctuating while **1** is bound.

On top of the A-loop conformational differences between these metastable states, other notable discrepancies are also evident. The conformation of the G-loop appears quite constant in the metastable state **1-$S_2$**, whereas there is more conformational freedom for this moiety in the other metastable states. The overall conformational stability in metastable state **1-$S_2$** is probably explained by the fact that there is only one dominant energy minimum in this state, whereas multiple ones exist for the other two states (Fig. 2a). Finally, a shift in the orientations of αC-helix exists as well between the states (residue 62–77 backbone RMSDs: 1.3–2.7 Å among all states; avg. 1.9 Å). For instance, in state **1-$S_2$** conformation of this helix is shifted up. Overall, these results highlight that p38α MAPK conformations are rather dynamic, when bound to **1**.

For **2**, MSM also revealed three metastable states (Fig. 3). On contrary to **1**, here one metastable state, **2-$S_3$**, is clearly dominant (85%), whereas the other two are rare (8% and 7% probability for **2-$S_1$** and **2-$S_2$**, respectively). Remarkably, even among these states, there are less readily observable conformational differences, which was not the case for metastable states related to **1**. With metastable states of p38α MAPK-**2**, the most notable conforma-tional differences are observed between G-loop configurations and in β-hairpin that is located close to the hinge region (Supplementary Fig. 8). Of note, the αC-helix conformation is relatively stable in all states (residue 62–77 backbone RMSDs: 1.1–1.9 Å among all states; avg. 1.5 Å). Clearly, the conforma-tional dynamics of p38α MAPK is stabilized when in complex with **2**, whereas the total opposite is observed with **1**, where

protein constantly fluctuates between different conformational states.

**Solvent exposure of the binding site residues and inhibitors.** The role of water is undoubtedly important in ligand binding[35]. Water competes with the ligand for target binding[36] and critically affects ligand dissociation[37]. For this reason, we next analysed its behavior in these p38α MAPK-inhibitor complexes.

Initially, we paid our attention to ligand–solvent contacts in the bound complexes. The solvent accessible surface area (SASA) appears slightly higher for **2** (Fig. 4a). Meanwhile, the solvent exposed polar surface area (formed by the polar N and O atoms) appear comparable for both inhibitors (Fig. 4b). In contrast, a clear discrepancy between the non-solvent exposed areas is observed between the inhibitors. Total buried surface area is clearly higher for the long residence time inhibitor **2** (Fig. 4c). Furthermore, this discrepancy is even more evident with the buried polar surface area of the inhibitors (Fig. 4d). These differences highlight that the long residence time inhibitor covers a larger area within its target which is not exposed to the solvent.

Next, we investigated whether the ligand affects the water exposure of the residues at the binding site. First, we evaluated the water exposure of the lipophilic residues of the protein within the binding site (Fig. 4d–j and Supplementary Fig. 9). Obviously, the most substantial difference is observed for water exposed Phe169. This residue is totally shielded from water with **2**, whereas with **1** it is fully exposed to water (median of ten water molecules within 3 Å). Moreover, Leu74 and Leu75 of the αC-helix are clearly less water exposed with **2**. However, not all lipophilic residues are better shielded from water when p38α MAPK is bound by the long residence time compound. For instance, Val38 and Leu171 are clearly less water exposed with **1**. The increased exposure of Val38 arises most likely from the shifted G-loop conformation with **2** and Leu171 is more shielded when it is in contact with dibenzosuberone scaffold of **1** (see Fig. 1e). Adjacent to these lipophilic residues within the binding site, there exist a triad of charged residues: the conserved lysine, Lys53, Glu71 of the αC-helix and Asp168 of the DFG-motif (Fig. 4k). With **2** the salt-bridge between the catalytic Lys53 is locked with the Glu71 of the αC-helix (99%) and rarely observed with Asp168 of the DFG-motif. With **1**, this balance is more shifted toward the Asp168 (88%). As we were curious about this difference and the role of solvent in here, we also checked solvent exposure of these acidic residues. Indeed, when the salt bridge with Lys53 is less stable, the sidechain of the acidic residue is more exposed to the solvent (Supplementary Fig. 10).

**Binding site resolution energy barrier is different for 1 and 2.** One key aspect in ligand dissociation is resolution of the binding site. This resolution may construct an energy barrier in the ligand dissociation[38]. Earlier, WaterMap[39,40] derived hydration sites energies with the docked p38α MAPK inhibitors were found to correlate with the observed binding kinetics[41]. Here, we eval-uated our MSM derived structures (conformations) and their binding pocket related hydration site energies by WaterMap, using the apo structures of these ligand bound configurations as an input for these calculations. The hydration sites overlapping with the bound inhibitor and their estimated energies indicate the energy barrier for the resolution of the binding pocket during ligand dissociation.

Based on the WaterMap calculations, hydration sites with highest energy for both compounds are located on the difluorophenyl moiety binding site in hydrophobic region I (HR I) (Fig. 5 and Supplementary Figs. 11 and 12). Also, dibenzosuberone scaffold is occupying high-energy hydration

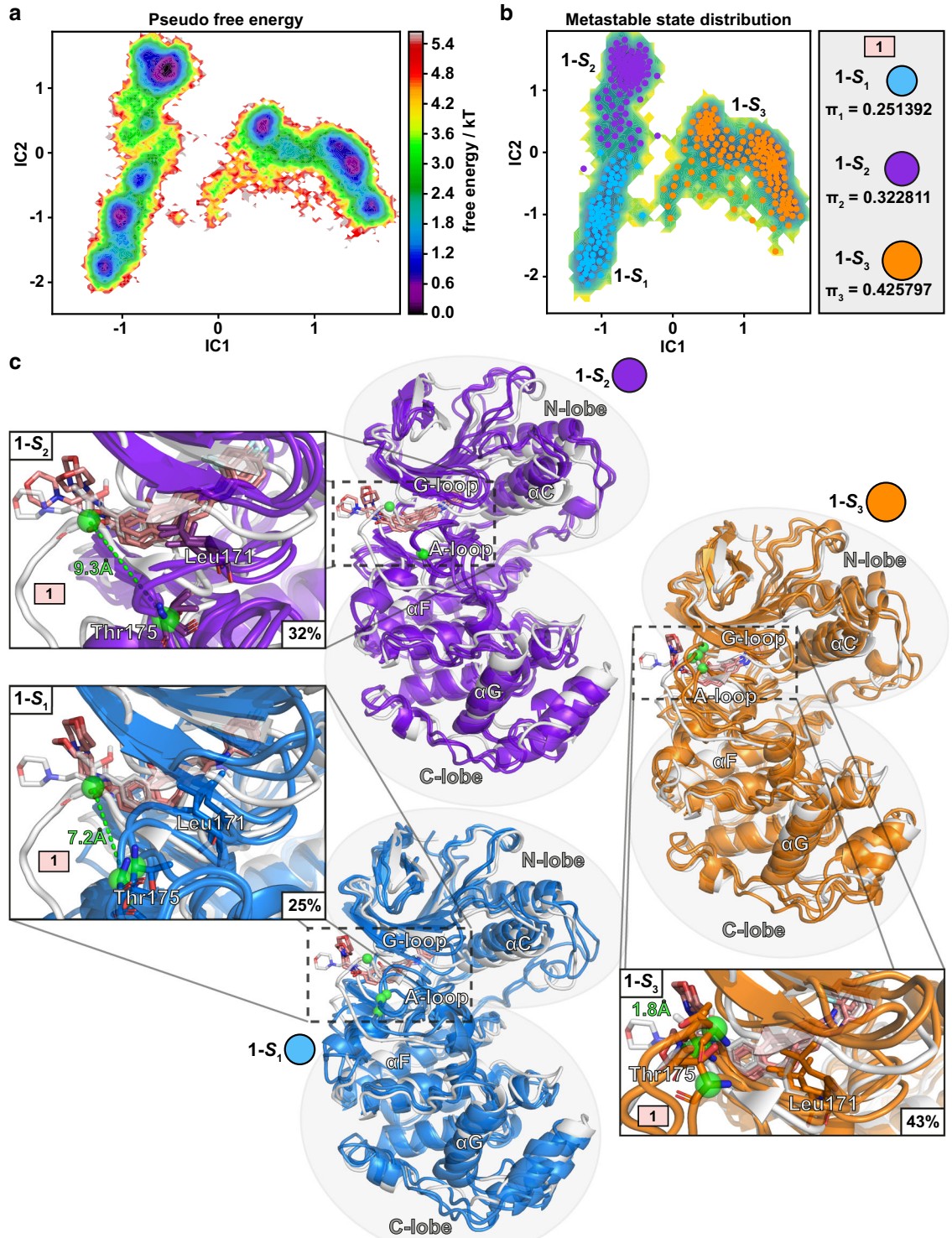

**Fig. 2 MSM metastable states of p38α MAPK-1 complex. a** Pseudo free energy map of distributions along time-lagged independent components (ICs) 1 and 2. **b** Separation of the three metastable states (**1-$S_1$**–**1-$S_3$**) by PCCA++ analysis. Equilibrium probability ($\pi_i$) for each state is indicated together with circles, which exhibit an area that is relative to their state probability. **c** Representative conformations of the metastable states. Each metastable state (**1-$S_i$**) is illustrated with three representative structures (colored cartoons: **1-$S_1$**, blue; **1-$S_2$**, purple; **1-$S_3$**, orange) and the simulation starting conformation is shown as a reference in gray. Locations of Thr175 Cα-atoms are highlighted with green spheres.

sites, whereas hydration site displacements related to the 2-morpholinoethylamide are generally less beneficial. Moreover, with type I½ moiety of **2**, lipophilic thiophene moiety displaces high-energy water molecule(s) whereas amide displaces generally more low-energy waters (Fig. 5c and Supplementary Fig. 12).

According to the WaterMap ligand scores (Fig. 5a), the metastable state **1-$S_1$** appears energetically most favorable for the short residence time compound **1** (in the context of the water molecule displacement). In this configuration, the binding pocket would be occupied by more high-energy waters and thus the

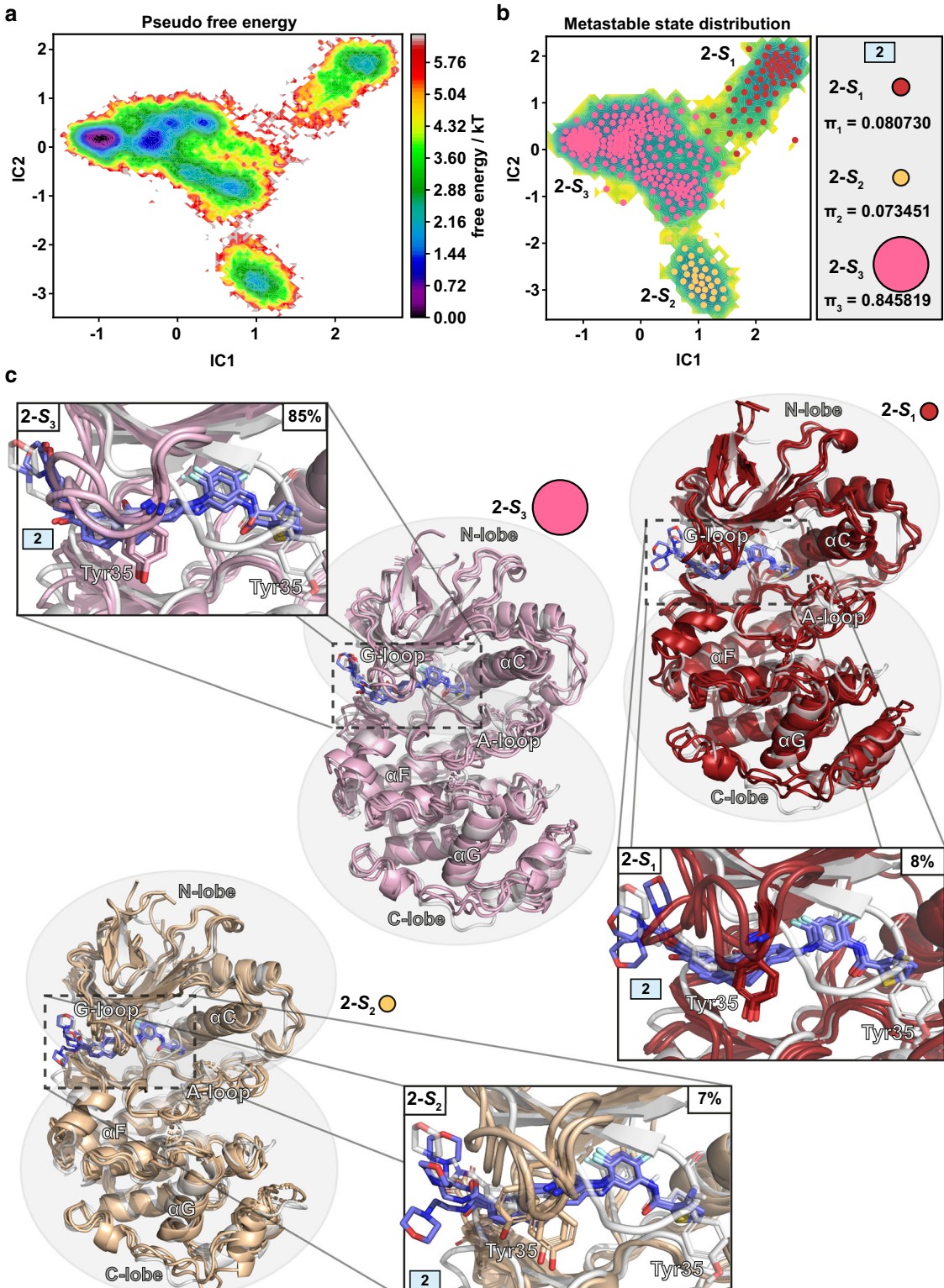

**Fig. 3 MSM metastable states of p38α MAPK-2 complex. a** Pseudo-free energy map of distributions along time-lagged independent components (ICs) 1 and 2. **b** Separation of the three metastable states (**2-S₁**–**2-S₃**) by PCCA++ analysis. Equilibrium probability ($\pi_i$) for each state is indicated together with circles, which exhibit an area that is relative to their state probability. **c** Representative conformations of the metastable states. Each metastable state (**2-S$_i$**) is illustrated with three representative structures (colored cartoons: **2-S₁**, red; **2-S₂**, wheat; **2-S₃**, pink) and the simulation starting conformation is shown as a reference in gray.

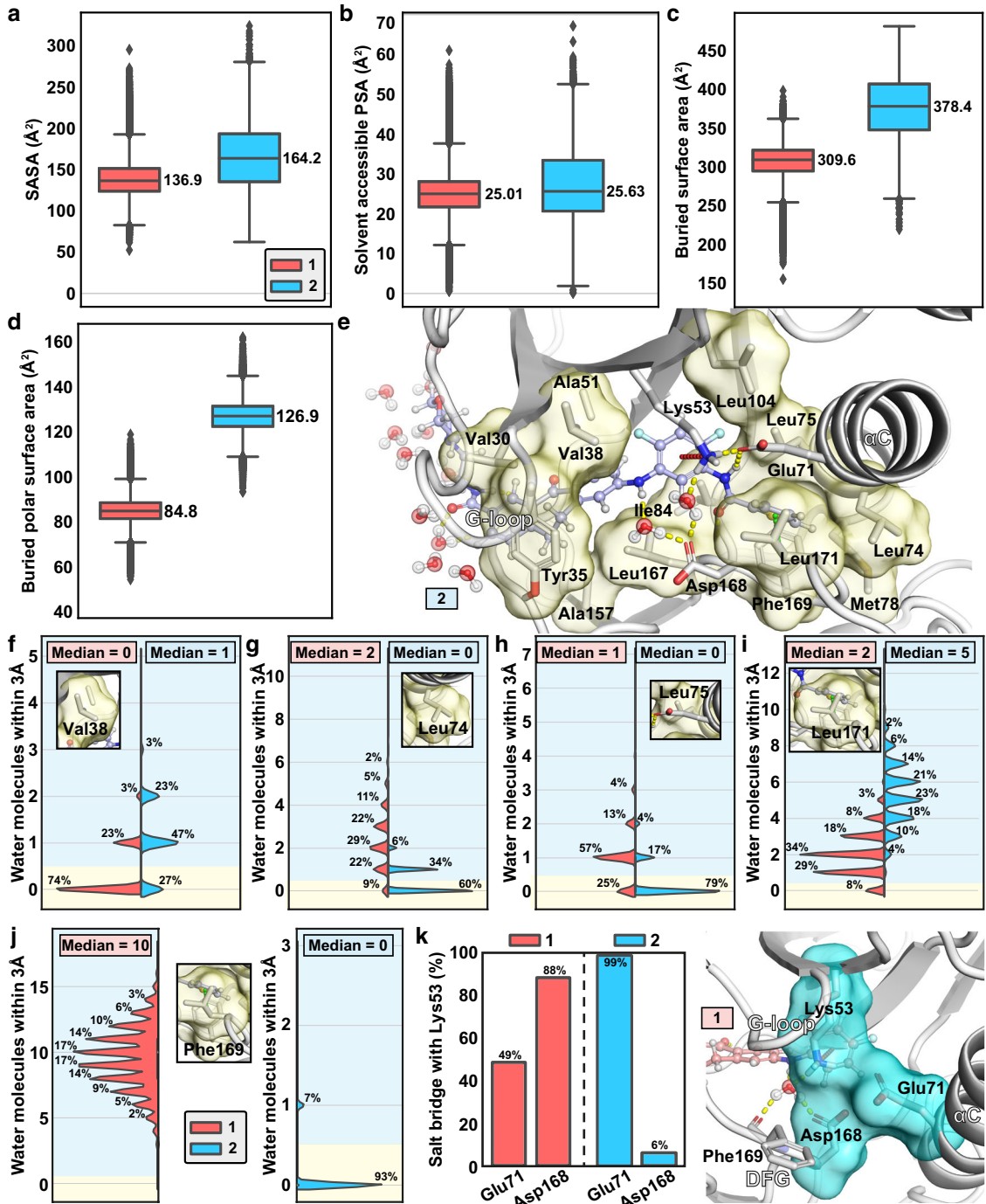

**Fig. 4 Solvent exposure of the inhibitors and the binding site residues. a** Solvent accessible surface area (SASA) of **1** and **2**. **b** Solvent accessible polar surface area of **1** and **2**. **c** Buried surface area (non-solvent exposed) of **1** and **2**. **d** Buried polar surface area of **1** and **2**. The black horizontal line in the box represents the median. Box displays the quartiles of the dataset (25–75%) and whiskers the rest of the data with maximum 1.5 IQR. Outliers are indicated with black diamonds. **e** Hydrophobic residues in the binding site that were monitored for their side chain solvent exposure are shown in stick model with their molecular surfaces illustrated in yellow. Compound **2** is shown in ball and stick model. Solvent exposure of Val38 (**f**), Leu74 (**g**), Leu75 (**h**), Leu171 (**i**), and Phe169 (**j**). **f–j** The blue background indicates solvent exposure (where ≥1 water molecule is located within 3 Å of the side chain) and the yellow background indicates when the side chain is not exposed to solvent. Solvent exposure of Val30, Tyr35, Ala51, Met78, Ile84, Leu104, Ala157, and Leu167 is shown in Supplementary Fig. 9. **k** Intra-protein salt bridge frequencies to catalytic Lys53. Simulation data was analysed for each 1 ns i.e., data in **a–k** consist of 91,328 and 86,505 individual data points for **1** and **2**, respectively. Source data are provided as a Source Data file.

resolution energy barrier of this configuration is the highest. However, with the more dominant metastable states **1**-$S_2$ and **1**-$S_3$ (together ~75%) less favorable water displacement by the ligand occurs ($\Delta G$ between −39.9 and −48.9 kcal/mol). This suggests a lower energy barrier to for resolution of the binding

pocket resolution in these protein configurations. The most consistent hydration site displacement energies among the structures derived from a single metastable state of **1** exist with state **1**-$S_2$. This is most likely related to the conformational stability (similarity) with the three representative structures of

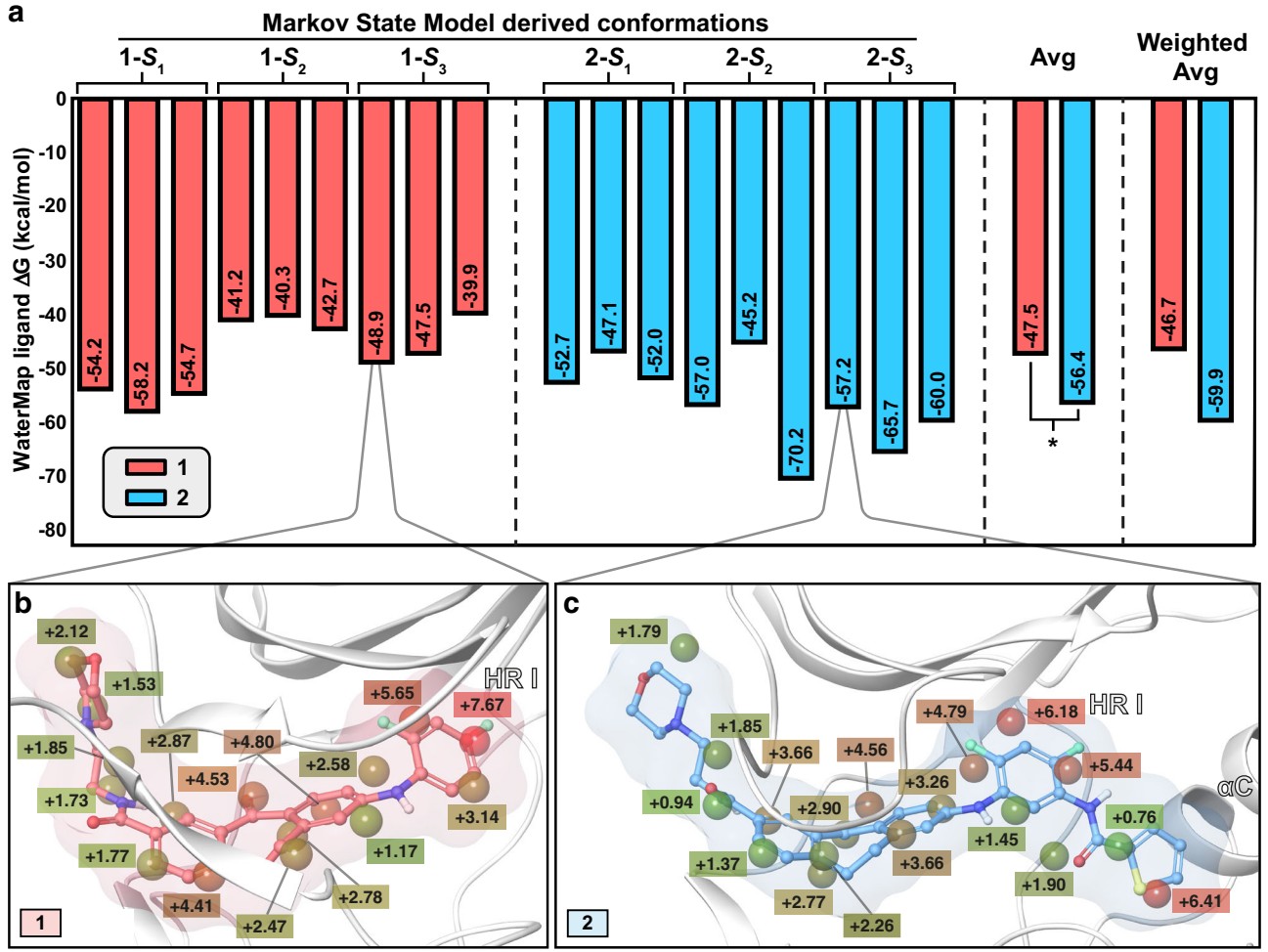

**Fig. 5 Higher energy barrier for resolution of the binding site is observed with 2. a** WaterMap ligand scores based on the ligand displaced waters in the MSM derived metastable state derived conformations. Three structures for each metastable state were evaluated. *$P < 0.05$ (two-tailed $t$-test). Weighted average is calculated by using the equilibrium probability ($\pi_i$) associated for each metastable state as the weight. **b** Displaced waters by **1** in the **1-$S_3$** conformation I (hydration sites with overlap factor >0.5 with the ligand are shown). **1** is shown with ball and stick representation (nonpolar hydrogens are hidden). **c** Displaced waters by **2** in the **2-$S_3$** conformation I (hydration sites with overlap factor >0.5 with the ligand are shown). **2** is shown with ball and stick representation (nonpolar hydrogens hidden). **b**, **c** The hydration sites are shown in spheres together with their estimated energies (kcal/mol, $\Delta G$ relative to bulk water) and are colored with the green-brown-red scale from low to high energy. Source data are provided as a Source Data file.

this state, especially on the G-loop (Fig. 2c). On average, **1** displays a score of −47.5 kcal/mol.

Generally higher energy hydration site displacement is observed with long residence time compound **2** (Fig. 5a). On average, **2** exhibits a score of −56.4 kcal/mol, which is significantly lower compared to what is observed for **1** (−47.5 kcal/mol). This difference is even more striking, if we consider the metastable state equilibrium probabilities ($\pi_i$) and use these as a weight for the energies for each state (weighted averages). Highest energy score observed for **2** (−45.2 kcal/mol) is related to the **2-$S_2$** structure, where the Tyr35 appears in a shifted configuration (Fig. 3c). This conformation probably allows easier solvation of the pocket (lower resolution energy barrier). Overall, a clear difference between the short and long residence time compounds exist between their water displacement energies (scores), which indicates a lower energy barrier for resolution of the binding site when **1** is bound to p38α MAPK.

**Stability of p38α MAPK associated with 2 is ligand-dependent.** As the two compounds prefer to bind different p38α MAPK conformations, we next set out to resolve what is the impact of the bound ligand to the protein behavior. To this end, we

changed the inhibitor with the other to observe ligand-dependent effects. As **2** is totally incompatible to the p38α MAPK-**1** conformation (steric clash), we were only able to simulate **1** in p38α MAPK-**2** associated protein configurations. For the start configuration we selected the dominant metastable state **2-$S_3$** and replaced the ligand with **1**. An aggregate of 90 µs simulations revealed that p38α MAPK behavior is indeed dependent on the bound ligand. First, the ultra-stable salt bridge between Lys53 and Glu71 is destabilized in this conformation with **1** (Fig. 6a). In fact, there seems to be a clear shift towards a similar salt bridge profile for Lys53 that was observed originally for **1** (Fig. 4k). Furthermore, the solvent exposure profile of the salt bridge residues and binding site hydrophobic residues is also shifted away from native profile of **2** (Fig. 6b–g). Especially, Phe169 appears highly water-exposed with **1** in this conformation. This highlights the key role of **2** in shielding this residue in this p38α MAPK conformation. Finally, we monitored the potential interactions between the kinase and **1** in this **2-$S_3$** conformation (Fig. 6h). Overall, **1** has lost or diminished its water-mediated interactions to the DFG residues Phe169 and Asp168. Moreover, it is unable to obtain as stable cation–π interaction to Lys53 (33%) as is observed for **2** (55%).

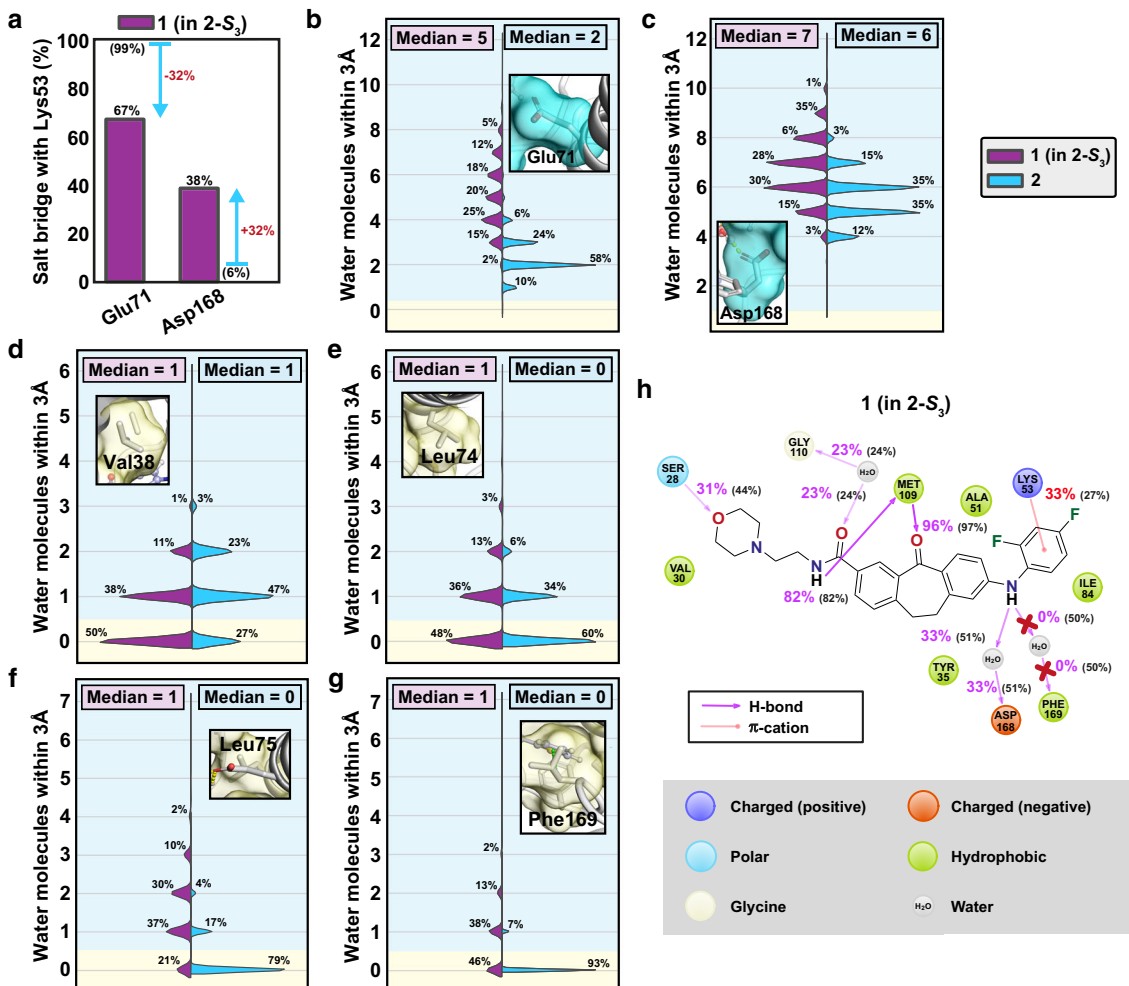

**Fig. 6 Changing the bound ligand in p38α MAPK–2 to 1 disrupts the stability of the kinase. a** Salt bridge profile of Lys53 with **1** in **2-$S_3$** is shifted towards the profile which is observed for p38α MAPK-**1**. Salt bridge interactions observed in simulations of **2** are shown in parenthesis and the differences are indicated with blue arrows. Solvent exposure of the salt bridge forming residues Glu71 (**b**) and Asp168 (**c**). Solvent exposure of the hydrophobic residues Val38 (**d**), Leu74 (**e**), Leu75 (**f**), and Phe169 (**g**). In **b-g** the blue background indicates solvent exposure (where ≥1 water molecule is located within 3 Å of the side chain) and the yellow background indicates when the side chain is not exposed to solvent. **h** Interactions for **1** when bound in **2-$S_3$** conformation. The inhibitor loses its key water mediated interactions to Phe169 and interactions to Asp168 are diminished. In parenthesis are the interaction frequencies observed for **1** in its preferred p38α MAPK binding conformation (Fig. 1d). Simulation data was analysed for each 1 ns i.e. data in **a–d** consist of 90,120 and 86,505 individual data points for **1** in **2-$S_3$** and **2**, respectively. Source data are provided as a Source Data file.

| Table 1 Observed ligand dissociation in well-tempered metadynamics simulations. | | | | | | |
|---|---|---|---|---|---|---|
| Ligand dissociation[a] | 1-$S_1$ | 1-$S_2$ | 1-$S_3$ | 2-$S_1$ | 2-$S_2$ | 2-$S_3$ |
| Yes | 59 | 54 | 53 | 8 | 5 | 9 |
| No | 1 | 6 | 7 | 52 | 55 | 51 |
| [a]Ligand was considered fully dissociated only if 15 Å distance was reached during the simulation. | | | | | | |

**Higher dissociation barrier of long residence time inhibitor.** To better understand if the force field's behavior with **1** and **2** is in line with the experimentally observed residence times, we studied the energy barrier for ligand dissociation from the binding site. To this end, we conducted well-tempered metadynamics simulations (see details in the "Methods" section). We ran 20 independent well-tempered metadynamics simulations for each MSM derived structure, which resulted in total of 60 simulations for each metastable state (Table 1 and Supplementary Figs. 13–30). Based on these simulations, a clear difference in the dissociation

energy barrier for the compounds **1** and **2** is evident. Dissociation of **1** from the binding site was associated with lower energy. While full dissociation was observed in 166 replicas of **1**, only in 14 replicas the ligand was maintained in the binding site. Conversely, **2** dissociated only in 22 out of 180 replicas, describing higher energy barrier for its dissociation. Overall, these results demonstrate an acceptable behavior for the force field that is in line with the experimental results.

**A short residence time inhibitor SB203580.** Finally, we compared how our findings translate to an inhibitor of a totally different chemical class. For this, we selected the widely studied **SB203580** (Fig. 7a), which displays a short residence time of 3.6 s in SPR and was even below the lowest limit of detection (LLOD) in our FP assay (Fig. 7b). Accordingly, it showed a distinct lower potency in the cell based MK2 translocation assay (Supplementary Figs. 1 and 2). Inhibitor stayed in the binding site throughout the conducted simulations (aggregate of ~90 μs), ensuring that the obtained simulation data is comparable to compounds **1** and **2**. **SB203580** displays stable H-bond interaction to Met109 (95%);

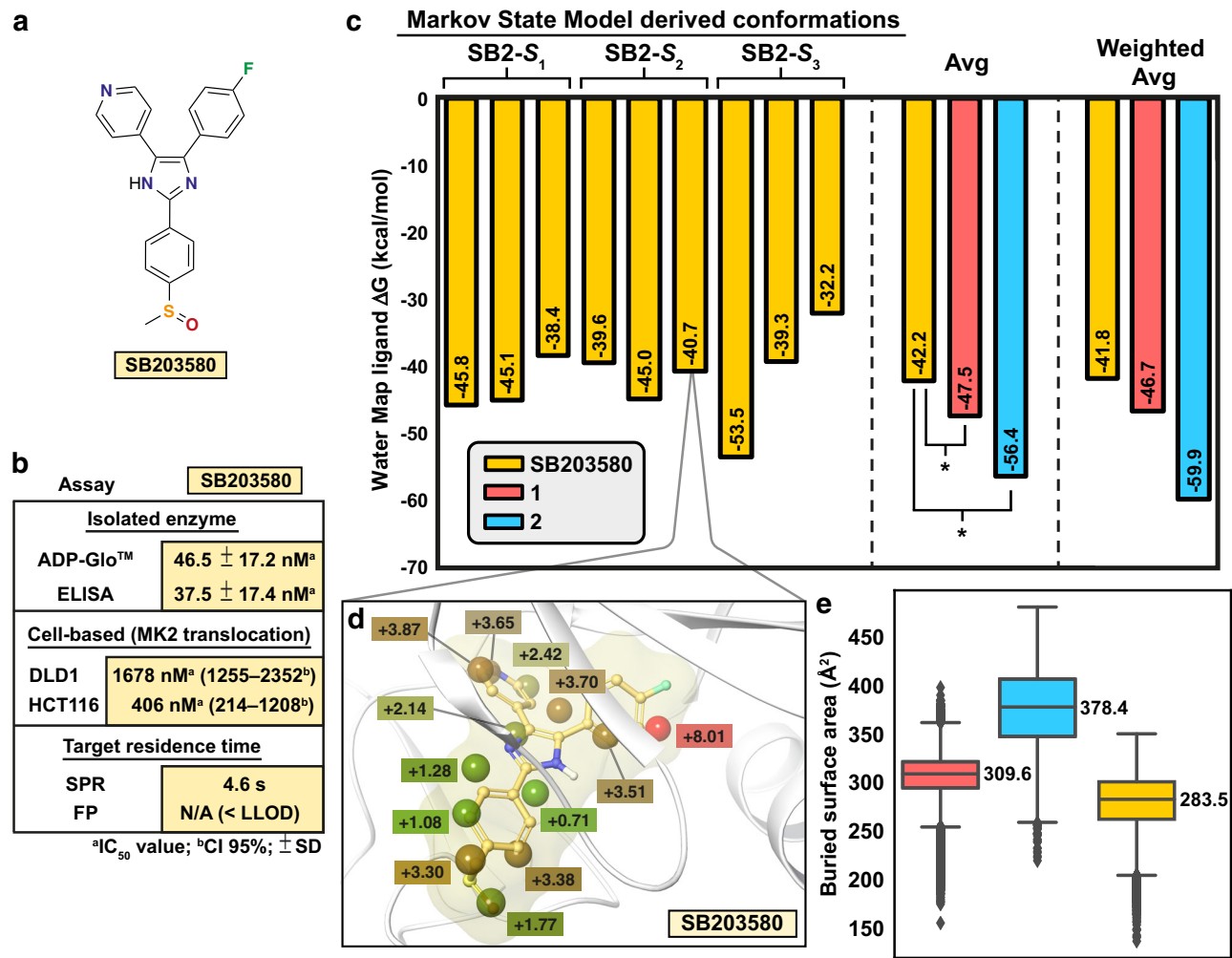

**Fig. 7 Short residence time inhibitor SB203580. a** Structure of **SB203580** is chemically diverse from **1** or **2**. **b** Biological activity of **SB203580** in selected assays. (ADP-Glo, ELISA, DLD1, HCT116, FP: $n = 3$; SPR $n = 2$) **c** WaterMap ligand scores based on the ligand displaced waters in the MSM derived metastable state derived conformations (see Fig. 6 for further details). *$P < 0.05$ (two-tailed $t$-test). **d** Displace**d** waters by **SB203580** in the **SB2-$S_2$** conformation II (hydration sites with overlap factor >0.5 with the ligand are shown). The hydration sites are shown in spheres together with their estimated energies (kcal/mol, $\Delta G$ relative to bulk water) and are colored with the green-brown-red scale from low to high energy. **SB203580** is shown with ball and stick representation (nonpolar hydrogens are hidden). **e** Buried surface areas of **1**, **2** and **SB203580**. Simulation data was analysed for each 1 ns i.e., data in **e** consist of 91,328, 86,505, and 91,090 individual data points for **1**, **2** and **SB203580**, respectively. The black horizontal line in the box represents the median. Box displays the quartiles of the dataset (25–75%) and whiskers the rest of the data with maximum 1.5 IQR. Outliers are indicated with black diamonds. Source data are provided as a Source Data file.

however, other interactions with p38α MAPK are less stable and thereby appear with lower frequencies (Supplementary Fig. 31). For instance, interaction between Phe169 and imidazole-ring appears with less than 20% frequency. Water mediated interaction and direct H-bond to Lys53 are displayed with 49% and 23% frequencies, respectively. While the fluorophenyl-ring of **SB203580** displays cation–π interaction to Lys53 with only 9% frequency, **1** and **2** exhibited comparable cation–π interaction with 27% and 51%, respectively. H-bond between sulphone oxygen and His174 (disordered in the crystal structure) appears with 29% frequency.

MSM analysis of this simulation data revealed that p38α MAPK conformational behavior in complex with **SB203580** appears more comparable to **1** than to **2** (Supplementary Fig. 32). **SB203580** also displays three states that appear with the probabilities of 64%, 28% and 8%. These states appear conformationally distinct, demonstrating the high conformational dynamics of p38α MAPK when in complex with **SB203580** (Supplementary Fig. 32). Conversely to **1**, **SB203580** needs

to adapt according to the protein motions in the binding site, as seen by its varying conformations in different structural states. Furthermore, this short residence time inhibitor exhibited smaller buried surface area compared to **1** and **2** (Fig. 7e). It also displayed occasionally considerably lower buried polar surface area (Supplementary Fig. 32). Finally, we conducted a similar WaterMap analysis of the MSM derived conformations as for **1** and **2**. Resolvation barrier for SB203580 appears the lowest among all studied kinase inhibitors (Fig. 7c and Supplementary Fig. 33). With an average of −42.2 kcal/mol and weighted average of −41.8 kcal/mol, resolvation barrier associated with **SB203580** is ~5 kcal/mol lower compared with **1**, and ~14–18 kcal/mol lower compared with **2**.

## Discussion

Here, we compared on the molecular level two structurally related inhibitors of p38α MAPK that display similar affinities but different residence times. Our unbiased long timescale MD

simulations revealed the key differences between these inhibitors when in complex with p38α MAPK. Even though the two studied compounds mainly exhibit similar interactions with p38α MAPK from their shared scaffold, remarkable differences were revealed in proteins' conformational stability and in solvent exposure of protein, together with the buried surface area of the ligand. Moreover, resolution of the ligand binding site appeared energetically more favorable for the inhibitor with short residence time. Further simulations with a structurally diverse a short residence time inhibitor **SB203580** corroborated these findings.

It has been shown earlier that protein conformational flexibility plays a crucial role in the residence time[42]. In the context of protein kinases, it was recently discovered that inhibitors that supported a more ordered configuration of the protein were associated with a longer residence time[43]. On the molecular level, explanations such as protein–ligand shape complementary[44] and steric constraints[45] have been suggested to be decisive. These steric effects are definitely related to the protein's conformational freedom and also to the solvent exposure of both compound and protein. Thus, agreeing with our observations here related to buried surface area of the ligand, protein conformational flexibility and solvent exposure of the binding site residues.

The short residence time inhibitor **1** had clearly lower amount of buried surface area compared to **2**. This non-solvent exposed area was also low with the ultra-short residence time inhibitor **SB203580**. This may also play an important role for residence time. Our MSM analysis revealed constant conformational fluctuations of p38α MAPK when in complex with **1**, which was also the case for **SB203580**, whereas with **2** the kinase appeared remarkably stable. Therefore, we anticipate that higher amount of buried surface area of a ligand is closely associated with lower conformational flexibility of the protein. This may be a result of less conformational strain arising from the water molecules accessing to these unfavorable high-energy water locations in the hydrophobic binding site, which are instead shielded by the ligand. Overall, this implies that increasing the non-solvent exposed area of a ligand could play a crucial role for the residence time. In practice, this could be perhaps addressed by aiming to find the best fitting compound to the desired stable protein configuration, which will then have the biggest buried surface area as possible for the particular conformation.

Regarding to the direct interactions to compounds' shared scaffold, we noticed one notable difference—the lipophilic dibenzosuberone moiety was shielded from the solvent by a different protein residue: Leu171 and Tyr35 for **1** and **2**, respectively. This difference appears to be related to the binding mode of the inhibitor (protein conformation). Interestingly, it seems that same principles apply here to this hinge binder scaffold regardless of the protein configuration. The lipophilic part of the ligand needs to be shielded from the solvent and this task is completed by a different but similar (hydrophobic) residue. Here, the remarkable flexibility and adaptability of p38α MAPK ensures that this is possible for this hydrophobic hinge binder in different protein conformations. This implies that the same kinase inhibitor scaffold (hinge binder) is useful among different protein configurations (with different type of kinase inhibitors) only if binding environment can adapt to the ligand requirements accordingly. Interestingly, together with **SB203580**, we observed that frequencies of cation–π interaction to Lys53 were ranked according to residence time of the compounds (the higher the interaction frequency the longer the residence time).

The MSM-derived metastable states revealed that the conformation of the αC-helix with **2** appears more stable compared with **1**, for which considerably shifted configurations of this helix occur. As described earlier, the type I½ inhibitors are believed to stabilize this helix via R-spine interactions. In addition to the

direct R-spine stabilization by **2**, we noticed that the salt bridge between Glu71 (from the αC-helix) and Lys53 was stabilized by this compound and was also accompanied by diminished solvent exposure. This was clearly dependent on **2**, as **1** was unable to stabilize this salt bridge in the p38α MAPK-**2** associated conformation. Diminishing the solvent exposure of this salt bridge potentially has an influence on its stability. For instance, it has been earlier demonstrated that shielding a salt bridge between ligand and protein from the solvent contributes to the binding[46]. This stabilized salt bridge interaction between Glu71 and Lys53 contributes to the enhanced conformational stability of the αC-helix. Furthermore, the stability of this salt bridge, which fixes Lys53's conformation, may be the reason for the enhanced cation–π interaction observed with compound **2** (51% vs. 27% with **1**).

Observations here related to the conformational dynamics in these complexes are in agreement with the available structural data. The starting structure used in simulations of **1**, p38α MAPK–Skepinone-L structure (PDB ID: 3que), was classified to belong into ωCD-class of structures[47]. This structural class was described to contain highly heterogenous structures that may represent transition states of the primary kinase conformations. Therefore, it is perhaps not surprising that such conformational fluctuation was also observed here in simulations of **1**. Analysis of the p38α MAPK-**2** structure (PDB ID: 5tbe) by KinaMetrix[48] classified it as CIDI (αC-helix-in, DFG-in), which configuration was stable throughout the simulations. SB203580 structure (PDB ID: 3gcp) is classified similarly to ωCD-class, which was noted also as dynamically resemble more **1** than **2**.

Water's role and importance in the ligand binding affinity is a well-established concept. Its role in the dissociation process, however, is generally less studied. To the best of our knowledge, evaluation of the hydration site energies related to solvent displacement by the ligand have never been combined with Markov State Model derived structures earlier. As these water energy evaluation methods such as WaterMap are usually extremely sensitive for subtle conformational changes in the protein, we anticipated that this combinatorial approach of the metastable state derived protein configurations may offer an improved averaged picture for the water energies related to the long time-scale conformational changes in the binding site. Using this approach, we observed significant difference between the ligand displaced hydration site energies. The energy barrier for the resolution of **2** bound p38α MAPK conformations is much higher compared to **1**, which is in agreement with the observed residence times. Furthermore, **SB203580** displayed the lowest resolution barrier, which from the studied inhibitors is also accompanied with the shortest residence time.

In summary, our results highlight the differences in protein conformational stability and amount of non-solvent exposed buried ligand surface area in the protein–ligand complex, improved protein–ligand interactions, along with the resolution energy barrier of the binding pocket between these structurally relevant inhibitors accompanied by similar affinities but different residence times. Consideration of these aspects already in the ligand design process is possible and could drive the compounds towards increased residence times that may result in the desired improvement in pharmacological efficacy.

## Methods

**ADP-Glo[TM]**. IC$_{50}$ determinations were performed using the luminescence-based ADP-Glo[TM] assay by Promega, which measures kinase activity based on ADP formation. All required reagents were prepared according to manufacturer's protocol. First, inhibitors were preincubated with p38α MAP kinase for 15 min. The kinase reaction was started by adding the substrate and ATP. After 60 min incubation time, the reaction was terminated by adding ADP-Glo[TM] reagent and an incubation time of 60 min. At last, ADP-Glo TM kinase detection reagent was

added and incubated for 30 min. All reactions were carried out on a plate shaker at room temperature. Final concentrations used for the kinase reactions were: 1.67 ng/µL p38α MAPK, 0.2 ng/µL substrate, and 150 µM ATP. The ratio between kinase reaction, ADP-Glo TM reagent and ADP-Glo TM kinase detection reagent was 1:1:2. The positive control consisted of p38α MAP kinase, substrate and buffer. The assay was performed in triplicates. The negative control consisted of all kinase reaction components without enzyme. Luminescence was detected using the plate reader FilterMax F5 by Molecular Devices with subsequent data analysis using GraphPad Prism version 7.0.0. for Windows (GraphPad Software, San Diego, CA, USA).

**Hotspot™**. Hotspot™ radiometric assay[49] was conducted by Reaction Biology (Reaction Biology Corp., Malvern, PA, USA). In brief, five-dose singleton measurements were conducted starting from 1 µM compound concentration with a ten-fold serial dilution (lowest concentration of 0.1 nM). ATP concentration of 10 µM was used in the assay.

**MK2 translocation assay**. For the cell-based in vitro translocation assay, HCT116 (CCL-247™, ATCC, Manassas, VA, USA) and DLD1 (CCL-221™, ATCC, Manassas, VA, USA) cell lines were stably transfected with a p38α MAP kinase specific reporter expression construct encoding a fluorescently labeled MAPKAP2 (eGFP-MK2)[50,51]. To study compound effects on p38α MAP kinase inhibition $1.2 \times 10^4$ cells/well were seeded into 96-well clear tissue-culture microplates (Bio-One µClear®, Greiner, Germany). HCT116 cells were cultivated for two days in DMEM medium (Gibco), DLD1 cells for two days in RPMI 1640 medium (Gibco), each substituted with 100 U/mL Penicillin and 100 µg/mL Streptomycin and grown until 80% confluence. Compounds (**1**, **2** and **SB203580**) were added to the growth medium to final concentrations of 10 µM–10 pM and incubated under cultivation conditions for additional 1 h. The p38 MAPK pathway was activated by hyper-osmolarity (addition of NaCl to 350 mM Osm) for 10 min inducing a nuclear to cytoplasmic translocation of effector kinase eGFP-MK2. For optical analysis, cells were fixed using 4% PFA/PBS following DAPI staining and subjected to automated high content analysis (HCA) image acquisition using an Image Xpress micro XL microscope (Molecular Devices). 40× Plan Apo objective was used to acquire FITC and DAPI signals at nine sites per well. MetaXpress® Custom Module Editor Software (64 bit, 6.2.3.733, Molecular Devices) was trained for automated image analysis. DAPI signal was used to define nuclear region of interest (ROI) and DAPI-negative cytoplasmic ring for respective cytoplasmic ROI. Non-cellular FITC background signals were subtracted from defined ROIs. Nuclear to cytoplasmic ratios were calculated from cellular fluorescence intensities of eGFP-MK2 reporter within defined ROIs. Cells with ratios within the 10–90 percentiles of all measured cells were taken into account. Ratios from 300 to 1000 cells per condition were averaged for each of three biological replicates, allowing reliable statistical measures and quantification. Z′ factors of 0.52 for HCT116_eGFP-MK2 cell line and 0.68 for DLD1_eGFP-MK2 cell line were calculated for the assay from non-stimulated and stimulated DMSO controls according to the formula: $z' = 1 - (3(\theta p + \theta n)/(\mu p - \mu n))$ $p$: positive control and $n$: negative control, $\theta$ standard deviation; $\mu$ mean[52].

**Fluorescence polarization**. The target residence time was determined by an adapted commercially available fluorescence polarization assay kit (Transcreener ADP by BellBrook Labs). First, the respective inhibitor was preincubated in 50 mM Tris [pH 7.5], 1 mM dithiothreitol, 10 mM MgCl₂, 10 mM β-glycerophosphate, and 0.1 mM Na₃VO₄ at a concentration of 20 * IC₅₀ with p38α MAPK (12.4 µg/mL) at room temperature for 1 h to enable the formation of an enzyme-inhibitor-complex. Next, the formed complex was "jump diluted" 1:100 (0.5 µL were diluted into 49.5 µL)in a 96 half area well plate with ATF2 (82 µM) as kinase substrate, ATP (40 µM) and ADP Kinase assay detection reagents. The plate was immediately measured every 3 min for 4 h. Fluorescence polarization was detected using the plate reader VictorNivo by PerkinElmer. The assay was performed in triplicates. The obtained data were analyzed via integrated rate equation in Graphpad Prism version 7.0.0. for Windows (GraphPad Software, San Diego, CA, USA). The residence time was subsequently calculated by the reciprocal value of $k_{off}$.

**Origin of compounds**. All inhibitors were synthesized in-house. The synthetic procedures and analytical characterization of compounds **1** and **2** have been published previously by our group[27,53] and are also provided here in detail in the Supplementary Methods. The compound **SB203580** was synthesized in our laboratories according to the procedures documented in patent literature (US5686455A)[54]. The analytical characterization is in agreement with the previously described data and can be found in the Supplementary Methods.

**Molecular modeling**. MD simulations, WaterMap simulations and all analysis were conducted with Maestro (versions 2018-4 and 2020-2) (Schrödinger, LLC, New York, NY, 2020). All modeling was conducted with OPLS3e[55,56] force field. Visualization was conducted with PyMOL (The PyMOL Molecular Graphics System, Version 2.2.3 Schrödinger, LLC).

**MD simulations**. For simulations of **1** we utilized the crystal structure of p38α MAPK in complex with Skepinone-L (PDB ID: 3que[29]), which is a highly similar ligand as **1**; thus, Skepinone-L was replaced by **1** to obtain p38α MAPK-**1** structure. For **2** we used the existing crystal structure (PDB ID: 5tbe[28]). Both complexes were prepared and energy minimized with Protein Preparation Wizard[57]. The missing sidechains and loops were added in this process with Prime[58,59]. For SB203580 we used the available co-crystal structure (PDB ID: 3gcp[60]). From the structure Glu4 was deleted and Leu353 was added to obtain similar system as with other compounds. Similar protein preparation was conducted as for other systems.

Simulations were conducted with Desmond[61] MD engine. Systems were then solvated in a 14 Å cubic box with TIP3P[62] water model together with 150 mM K⁺ and Cl⁻ ions (adjusted to neutral net charge). The final systems comprised ~65k atoms. Prior to the production simulations, the default Desmond relaxation was applied for both systems. The production simulations were run in NpT ensemble (Nosé–Hoover method; $p = 1.01325$ bar, Martyna–Tobias–Klein method, $T = 310$ K) with default settings. RESPA integrator with 2, 2, and 6 fs timesteps were used for bonded, near and far, respectively. For Coulombic cutoff, the default value of 9 Å was used.

For all systems, first an initial 10 µs simulations were run. From these initial simulations further simulations were derived from different protein configurations to increase the sampling, finally achieving ~91 µs simulation data for **1** and ~86 µs for compound **2** (see details of simulations in Supplementary Fig. 34). **SB203580** simulations, were conducted in a similar manner resulting in an aggregate of ~91 µs of simulation data (Supplementary Fig. 35). Simulations of **1** in **2-S₃** metastable state conformations I–III were generated by replacing the **2** with **1** in the complexes and the systems were prepared similarly as other systems, except in the restrained energy minimization a higher 0.60 Å RMSD convergence of the heavy atoms was applied. In total, ten replicates for each of the three conformations were simulated with the simulation length of 3 µs, which resulted in total of 90 µs simulation data (Supplementary Fig. 35).

**MD simulation protein–ligand interaction analysis**. Protein–ligand interactions were analysed by Simulation Interaction Analysis tool (Schrödinger LLC), with the following default definitions: hydrogen bonds: distance of 2.5 Å between the donor and acceptor with ≥120° and ≥90° for donor and acceptor angles, respectively; π–cation interactions: 4.5 Å distance between the positively charged and aromatic group; π–π interactions: stacking of two aromatic groups face-to-face or face-to-edge; water bridges: distance of 2.8 Å between the donor and acceptor with ≥110° and ≥90° for donor and acceptor angles.

**Markov state modeling**. Bayesian MSMs (individual models for compounds **1**, **2**, and SB203580), were built with PyEMMA2 using general recommendations[63]. As an input, trajectories containing only protein and ligand were included. For MSM of compound **1**, sidechain of K118 was omitted and for compound **2**, sidechain of K118 and residues A172, R186, I250, and E253 were not included. For data featurization, backbone torsions were used, with VAMP-2 scores[64] of 1.44, 2.01, and 1.89 for MSM of compound **1**, **2**, and SB203580 respectively. Time-lagged independent component analysis (TICA)[65] was utilized for dimension reduction of the data. For both models lag time (τ) was set for 40 ns, and two dimensions were selected. TICA output was clustered using k-means clustering, with √n for the number of clusters implied, 301 clusters for compound **1** and SB203580, and 293 clusters for **2**. Implied timescales were converged at the selected lag time (Supplementary Figs. 36 and 37). Microstates were grouped in three metastable states by the spectral clustering with Perron-cluster cluster analysis (PCCA++)[66]. Chapman–Kolmogorov validation test demonstrated an acceptable three state models for both systems (Supplementary Figs. 36 and 37).

**Ligand surface analysis**. The ligand's solvent accessible surface area (SASA) (Å²), polar surface area (PSA) (Å²), molecular surface area (MolSA) (Å²), and solvent accessible polar surface area (SAPSA) (Å²) were calculated by event_analysis.py and analyze_simulation.py scripts (Schrödinger LLC) using 1 ns interval in the analysis. A probe with the size of 1.4 Å radius is applied to calculate the surface area. With PSA (and SAPSA), only surface area that is contributed by N or O atoms is considered. The buried surface area was calculated by reducing the SASA from the MolSA. The buried polar surface area was calculated by reducing the SAPSA from the PSA.

**Protein solvent exposure analysis**. The solvent location next to the individual protein sidechains were analysed by trajectory_asl_monitor.py script (Schrödinger LLC). Total number of observed water molecules within 3 Å of the selected sidechain were monitored, using 1 ns interval in the analysis.

**WaterMap**. MSM derived structures were prepared with Protein Preparation Wizard[57]. For these energy-minimized structures, WaterMap[39,40] simulations, where non-solvent heavy atoms are restrained, were run using default settings with 2.0 ns simulation time with TIP4P water model. Waters within 10.0 Å from the ligand were included in the analysis. As the MSM derived structures did not include water molecules, all hydration site information was solely based on the WaterMap simulation.

**Well-tempered metadynamics simulations**. For the MSM derived structures, which were solvated in a box of minimum distance of 15 Å from the protein (KCl and TIP3P as in other simulations), 200 ns well-tempered Desmond metadynamics simulations with 20 replicas for each of the three metastable state derived structures were conducted ($20 \times 3 \times 3 = 180$ simulations). As a collective variable, we applied the distance of the mass centers of the ligand and binding site residues: Val38, Ala51, Leu75, Ile84, Thr106, Met109, with the width set to 0.03 Å and wall to 35 Å. Height of 0.2 kcal/mol with 1.0 ps interval and kTemp of 3.4 was used in the simulations; thereby, applying a similar bias in both systems.

**Reporting summary**. Further information on research design is available in the Nature Research Reporting Summary linked to this article.

## Data availability

The original Desmond raw-trajectories, PDB-coordinates for the energy minimized metastable state derived structures and raw-trajectories of the well-tempered metadynamics simulations generated and analysed during the current study have been deposited in the Zenodo repository and are freely available at: https://doi.org/10.5281/zenodo.4568113 (compound **1**); https://doi.org/10.5281/zenodo.4572444 (compound **1**); https://doi.org/10.5281/zenodo.4561797 (compound **2**); https://doi.org/10.5281/zenodo.4563896 (compound **2**); https://doi.org/10.5281/zenodo.5563359 (**SB203580**); https://doi.org/10.5281/zenodo.5563655 (**SB203580**); https://doi.org/10.5281/zenodo.5564118 (compound **1** simulated in compound **2** metastable state **2**-*S*₃); https://doi.org/10.5281/zenodo.5564208 (compound **1** simulated in compound **2** metastable state **2**-*S*₃); https://doi.org/10.5281/zenodo.5564586 (well-tempered metadynamics simulations of compounds **1** and **2**); https://doi.org/10.5281/zenodo.5570882 (well-tempered metadynamics simulations of compounds **1** and **2**); https://doi.org/10.5281/zenodo.5571352 (well-tempered metadynamics simulations of compounds **1** and **2**). Other data generated in this study are provided in the Supplementary Information and Source data file. The protein structures from the Protein Data Bank are available under the entry numbers 3que, 5tbe, and 6zqs. Source data are provided with this paper.

## Code availability

Jupyter notebooks used for MSM generation with PyEMMA2[63] are freely available at https://doi.org/10.5281/zenodo.5770578.

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

## Acknowledgements

The authors wish to acknowledge CSC—IT Center for Science, Finland, for computational resources. T.P. acknowledges the European Union's Horizon 2020 research and innovation programme (Marie Sklodowska-Curie grant agreement No 839230) and Orion Research Foundation sr for funding. S.L. and iFIT are funded by the Deutsche Forschungsgemeinschaft (DFG, German Research Foundation) under Germany's Excellence Strategy—EXC 2180—390900677. TüCAD2 is funded by the Federal Ministry of Education and Research (BMBF) and the Baden-Württemberg Ministry of Science as part of the Excellence Strategy of the German Federal and State Governments.

## Author contributions

T.P. and S.L. designed the study. T.P. performed and analysed the MD simulations. P.K and M.K performed the biological experiments. T.P drafted the manuscript; T.P., P.K., M.K., M.F., U.R., and S.L. revised the manuscript.

## Funding

## Competing interests

The authors declare no competing interests.
