## [Peer Review File · Nature Communications]

Decisive Role of Water and Protein Dynamics in Residence Time of p38 α MAP Kinase InhibitorsEditorial Note: Parts of this peer review file have been redacted as indicated to maintain the confidentiality of unpublished data.

REVIEWER COMMENTS

Reviewer #1 (Remarks to the Author):

The manuscript “Decisive Role of Water and Protein Dynamics in Residence Time of p38 α MAP Kinase Inhibitors” describes an in-depth computational analysis of two inhibitors of p38 α that differ by their residence time. Potentially, such analyses are very relevant for drug design processes, as they provide vignettes of different effects that play a role.

While I find that the methods have been executed to high standards of the field (with exceptions noted below), I am not convinced that the results can justify the conclusions. My main reservations center around the fact that only two ligands were analyzed: while the results certainly seem to correlate with the experimental evidence, there is a high risk that this correlation happened by chance. Moreover, correlation does not imply causation, so at the very minimum, a third ligand needs to be analyzed now that the hypotheses have been formulated and it needs to be shown that the behavior of this third ligand (which should either be a short or a long residence time ligand and not only a trivial modification of one of the two ligands already discussed) is consistent with expectations (for an additional caveat, see below).

In more detail, I would like to provide the following comments:

* As protein stability is a major difference claimed by the authors, it is concerning that the simulations started from protein structures in quite different conformations (states of activation). Hence, it is not clear that the observed difference in stability is only due to the ligands, it might also be a function of completely different basins of the energy landscape being sampled (in particular since unbiased simulations were used). I would be more convinced if the authors could demonstrate that a conformation very close to starting structure 1 is found in the trajectory of ligand 2 and vice versa. Even better, the authors would conduct simulations starting from the mixed complexes (i.e. ligand 1 in 5tbe and ligand 2 in 3que) and show that the results are not largely due to the protein conformation. This would be an appropriate negative control in my view.

* The experimental values in Fig. 1 are given without any error values. Neither are the numbers of independent repetitions mentioned in the Methods. Only in the case of the HotSpot experiment, the number of technical replicates is given as 1 (which is a concern in itself). While I do not doubt the trends observable in the data, the magnitudes might differ considerably. The authors already state this themselves, as the residence time difference shrinks quite a bit when changing experimental setup. This is not unusual for different measurement principles, but begs the question how big the difference between the ligands really are. Moreover, affinities are given as IC₅₀s, which are inherently incomparable. Thus, the statement that both ligands bind with the same affinity is not

entirely reproducible. Either the authors should convert to K_i values or at least ascertain that the compounds were measured on the same day in the same batch of experiments.

* In my opinion, the authors should adapt the strength of their claims (“decisive role”, “extremely stable”, etc) to the strength of their arguments (see above). For the “extremely stable” statement, I’d like to see RMSD or RMSF plots, for instance, to judge for myself.

* The manuscript falls somewhat short of the claim made in the Introduction “We anticipated that a better understanding of this process can provide general principles, which could be applied in the compound design process when aiming for improved TRT.”. Assuming that the factors identified also hold true under the additional analyses mentioned above, what are tangible design recommendations to improve these particular ligands? Ideally, the authors would translate this into a novel ligand with ultra-long TRT themselves, but I realize that this is beyond the scope of this manuscript (but not beyond the abilities of the authors!).

* Personally, I find the structural figures quite crowded and hard to see what the authors intend to show me. Less information might be more here.

Reviewer #3 (Remarks to the Author):

The authors use molecular dynamics simulations to try to understand why the residence times of two structurally similar MAPK inhibitors differ dramatically. They provide a number of though provoking hypotheses that are of great importance and general interest. However, significant additional work is needed to make the arguments compelling.

My primary concern is whether the simulation results are anything more than correlative. Protein force fields have come a long way, but the reliability of the parameters one can find for arbitrary chemical compounds is questionable. Therefore, it is essential that the authors demonstrate that their simulations are consistent with experiments before trying to draw any conclusions from them. The most obvious candidate from my perspective is to measure the dissociation time computationally, and see if the rates are similar to those in experiment (at a minimum the simulations have to give the same rank order). The absolute timescales are far beyond reach of conventional MD simulations. However, there are a variety of methods that are likely up to the task and where code is freely available on github. I am specifically thinking of our own FAST adaptive sampling method, Alex Dickson’s weighted ensembles approach, and Pratyush Tiwary’s methods.

My second major issue is the indirect nature of the comparison between the two complexes. Separate MSMs are built for the two different complexes, which makes direct comparison difficult. It would be interesting to build a common state space for the two datasets (e.g. define the states based on the union of the two datasets) and then estimate separate MSMs for each dataset so that the authors can directly assess which states are more/less populated in each case. This is an issue throughout the paper.

I also have a few minor issues that should be easier to address.

How were representative structures selected for each state? Given the authors coarse-grain extensively (to only 3 states), it seems like there is probably substantial conformational heterogeneity within each state. Building higher resolution MSMs with more states would be informative.

I also strongly suggest that the authors minimize the use of unnecessary acronyms, like TRT and cmpds.

It is unclear how the structures shown in Figure 1 were selected. No statistical evidence is given that they are representative.

Finally, I would like to (partly selfishly) suggest the following review article be cited for the use of MSMs to address problems like this one:

Advanced Methods for Accessing Protein Shape-Shifting Present New Therapeutic Opportunities

Reviewer #2 (Remarks to the Author):

This manuscript by Pantzar and co-authors used mainly computational methods to study an interesting system. The two similar compounds, 1 and 2, bind to p38a MAP kinase with the same K_d but very different kinetic behavior. The Laufer group has been studying this scaffold for a few years. Interestingly, this compound 2 has an additional function group that makes it become another type of inhibitors, and the group termed it type 1 ½ inhibitor. While there are experimental structures and biochemical assays, the data cannot explain why the two have the same binding affinity but compound 2 has much slower k_{off} (also k_{on}). This study therefore carried out very long classical molecular dynamics simulations for the two complex systems and investigate protein dynamics and the role of water molecules. They also include some experimental results. The study about conformational differences of the MAPK when it binds to the two similar inhibitors is quite interesting. The analysis of water molecules is informative, too. The work utilized a commercial package, and the computation setup was reasonable. While the results are reliable, the current interpretation about conformational dynamics and long residence time (line 160) is not accurate and need to carefully revise.

It is very interesting that the results from SPR and FP are quite different. The measured residence time of Compound 2 (slower one) becomes faster when using FP, but the residence time of compound 1 (fast) becomes 10 times slower. Did the authors see similar trends for other inhibitors? Although it may be out of the scope of the paper, the authors may discuss more about it. Why does the presence of ATP slow down compound 1 unbinding so much? Is it possible that the discrepancy between SPR and FP is because of use of the flowing buffer in SPR?

The simulations of only the bound states showed that the small chemical differences can lead to very different protein conformations. However, the reviewer cannot clearly see these differences from their figure 1 or other figures. Some figures that super-impose representative conformations or local energy minima from the two complexes should be helpful. This may also further explain why the water analysis using local energy minima showed different resolution energy.

The entire simulations only sampled the inhibitor bound state, which are also pretty similar to their crystal structure bound conformations. Therefore, the major effort of this study is trying to get clues from the bound complex (see the copy/paste figure from the ref #8, lower right energy minima) to understand binding kinetics. It is not easy because the kinetic property is largely determined by the intermediate (transition) states. This simplified figure from their Ref #8 can be used to present their experiments that compound 2 (red solid line, longer residence time) should have larger energy barrier than that of compound 1 (black line).

[redacted]

Since the simulations are for the bound state, the conformational dynamics near the bottom of the energy well does not directly correlate to the energy barrier. Instead, based on thermodynamics, $G = H - ST$, if the bound state has more stable local energy minima, it may reduce entropic loss during binding. Even the constantly fluctuating conformations of the compound 1 complex might result in less favorable binding enthalpy, considering entropy, the system is not less stable. The data or thermodynamics/kinetics theories do not support that more equally stable local energy minima (actually they only showed 3 here which is very few) directly correlate to lower free energy barrier, or one dominant energy minima in the bound state means larger free energy barrier. The authors concluded that 3 more equally populated minima (compound1, shorter residence time) vs 1 dominate minima (comp 2, longer residence time) in the bound state was too preliminary. Fig 2A and 3A cannot support their statements, and even the authors report the MSM kinetics info between those local energy minima is still useless. If they read their cited ref #41, that paper suggested that slower dissociation ligand has entropy driving binding and are more flexible, see their Abstract “Compounds bound to the helical conformation display slow association and dissociation rates, high-affinity and high cellular efficacy, and predominantly entropically driven binding. An important entropic contribution comes from the greater flexibility “. Therefore, while the resolution free energy approximation can be reasonably corrected to the free energy barrier, the very few local energy minima cannot bring any reasonable conclusion about the kinetic property.

Minor:

Page 4, line 76, the authors introduced the two inhibitors in this study. Their compound 1 is type I and compound 2 is type 1 ½ inhibitor, correct? The authors didn't mention it in the Abstract, and it may be better to at least mention it in Introduction.

Compared to the previous version, we have now more than doubled the simulation data in our revised manuscript (now in total of ~360 μ s plus 72 μ s of metadynamics simulations). The new simulations include simulations of a new structurally diverse short residence time inhibitor SB203580 and simulations of compound **1** in the dominant metastable state derived conformations of compound **2**. In addition, we have included well-tempered metadynamics simulations for the MSM derived structures of both compounds **1** and **2** (total of 360 simulations of 200 ns each).

We noted an erroneous calculation in our previous solvent accessible polar surface area (based on a typo in the software); therefore, we have removed the ASolvE values from the manuscript. However, these are now replaced by new surface area calculations, including both solvent accessible and buried surface areas. These new data may cope with the reviewers concerns.

Below you will find our detailed point-by-point response to the reviewers' concerns.

Reviewer #1 (Remarks to the Author):

The manuscript "Decisive Role of Water and Protein Dynamics in Residence Time of p38 α MAP Kinase Inhibitors" describes an in-depth computational analysis of two inhibitors of p38 α that differ by their residence time. Potentially, such analyses are very relevant for drug design processes, as they provide vignettes of different effects that play a role.

While I find that the methods have been executed to high standards of the field (with exceptions noted below), I am not convinced that the results can justify the conclusions. My main reservations center around the fact that only two ligands were analyzed: while the results certainly seem to correlate with the experimental evidence, there is a high risk that this

correlation happened by chance. Moreover, correlation does not imply causation, so at the very minimum, a third ligand needs to be analyzed now that the hypotheses have been formulated and it needs to be shown that the behavior of this third ligand (which should either be a short or a long residence time ligand and not only a trivial modification of one of the two ligands already discussed) is consistent with expectations (for an additional caveat, see below).

We thank the reviewer for pointing out the risk of chance in our results and totally agree that one should always be cautious with potential arbitrariness in correlation-causation relationships. As the reviewer suggested, we have now included a third ligand to the study, the well-known and most cited p38 α MAPK inhibitor SB203580. This inhibitor belongs to a totally different chemical class compared to our in-house compounds and represents a ligand accompanied with a very short residence time. A similar set of simulations were conducted for this ligand. Indeed, most of our key findings seem to translate to this inhibitor as well. It is less potent in the biological assay, reflecting to its ultra-short residence time (SPR = 4.6 s, and <LOOD in our FP assay). Based on MSM and interaction analysis, protein is more flexible, and the ligand displays lower buried surface areas (buried surface area and buried polar surface area). Especially the lower resolvation energy stands out from the data, which we have now included with a new main figure (**Figure 7**). Interestingly, also among the three compounds the cation– π interaction frequencies to Lys53 appeared to be ranked according to the residence time (the higher the interaction frequency the longer the residence time).

Overall, our findings seem to translate also to a structurally diverse inhibitor.

Figure 7. **Short residence time inhibitor SB203580.** **a** Structure of **SB203580** is chemically diverse from **1** or **2**. **b** Biological activity of **SB203580** in selected assays. **c** WaterMap ligand scores based on the ligand displaced waters in the MSM derived metastable state derived conformations (see Figure 6 for further details). * $P < 0.05$ (two-tailed t-test). **d** Displaced waters by **SB203580** in the **SB2-S₂** conformation II (hydration sites with overlap factor > 0.5 with the ligand are shown). The hydration sites are shown in spheres together with their estimated energies (kcal/mol, \otimes G relative to bulk water) and are coloured with the green-brown-red scale from low to high energy. **e** Buried surface areas of **1**, **2** and **SB203580**. Simulation data was analysed for each 1 ns i.e. data in **D** consist of 91,328, 86,505 and 91,090 individual data points for **1**, **2** and **SB203580**, respectively. The black horizontal line in the box represents the median. Box displays the quartiles of the dataset (25%–75%) and whiskers the rest of the data with maximum 1.5 IQR. Outliers are indicated with black diamonds.

In more detail, I would like to provide the following comments:

* As protein stability is a major difference claimed by the authors, it is concerning that the simulations started from protein structures in quite different conformations (states of

activation). Hence, it is not clear that the observed difference in stability is only due to the ligands, it might also be a function of completely different basins of the energy landscape being sampled (in particular since unbiased simulations were used). I would be more convinced if the authors could demonstrate that a conformation very close to starting structure 1 is found in the trajectory of ligand 2 and vice versa. Even better, the authors would conduct simulations starting from the mixed complexes (i.e. ligand 1 in 5tbe and ligand 2 in 3que) and show that the results are not largely due to the protein conformation. This would be an appropriate negative control in my view.

We thank the reviewer for this again very valuable comment. However, we have clear experimental evidence (based on the crystal structures of many inhibitors of the Skepinone-L derived series) that the binding conformations between these inhibitors are different. We have now included a new SI Figure (**Supplementary Figure S4**) to demonstrate the key differences in the compounds binding mode (please see also the response to the reviewer #2). Moreover, there is no interconversion between the conformations or no similar protein conformations for the ligands in the simulations, which is perhaps not surprising as these studied inhibitors do clearly prefer distinct protein configurations. In fact, we suppose that this is one of the key aspects for the longer residence time associated with type I_{1/2} inhibitor and we set out to resolve what is associated with these conformations in the dynamic environment (beyond the static conformations). Nevertheless, as we found this idea suggested by the reviewer very interesting, we have now implemented it indirectly into the study. We have now included new simulations (~ 90 μ s) where we have exchanged compound **1** and **2** in their most populated MSM derived conformations and studied the stability of the ligand/protein in this non-preferred bound state conformation. Obviously, this was doable only in one way, as compound **2** is unable to bind to a compound **1** conformation (**2** is unable

to fit due to steric constraints into the binding site in the **1** related protein conformation). This new data describes the behavior of **1** in the protein configuration only associated to **2**. When **1** is simulated in compound **2** associated p38 α MAPK conformation, we observe a clear destabilization on the ultra-stable Lys53–Glu71 salt bridge (99% frequency in simulations of **2**), which is shifted towards the native profile observed for **1**. Furthermore, the solvent exposure of the binding site residues is changed, highlighting the important shielding role of **2** in this p38 α MAPK conformation. Moreover, **1** is incapable to form as stable cation– π interaction to Lys53 as observed with **2**. A new section with a main figure (**Figure 6**) is now included to the manuscript to describe these results. Finally, we would like to point out related to the reviewer's worry about the impact of the protein conformation that the new compound **SB203580** binds to a distinct conformation compared to compounds **1** and **2**. Therefore, as the new results for this are in line with our earlier findings, we can conclude that the results are not largely due to the protein conformation.

Figure 6. Changing the bound ligand in p38 α MAPK-2 to 1 disrupts the stability of the kinase. **a** Salt bridge profile of Lys53 with 1 in 2-S₃ is shifted towards the profile which is observed for p38 α MAPK-1. Salt bridge interactions observed in simulations of 2 are shown in parenthesis and the differences are indicated with blue arrows. Solvent exposure of the salt bridge forming residues Glu71 (**b**) and Asp168 (**c**). Solvent exposure of the hydrophobic residues Val38 (**d**), Leu74 (**e**), Leu75 (**f**) and Phe169 (**g**). **h** Interactions for 1 when bound in 2-S₃ conformation. The inhibitor loses its key water mediated interactions to Phe169 and interactions to Asp168 are diminished. In parenthesis are the interaction frequencies observed for 1 in its preferred p38 α MAPK binding conformation (**Figure 1d**). Simulation data was analysed for each 1 ns i.e. data in **a-d** consist of 90,120 and 86,505 individual data points for 1 in 2-S₃ and 2, respectively.

* The experimental values in Fig. 1 are given without any error values. Neither are the numbers of independent repetitions mentioned in the Methods. Only in the case of the HotSpot experiment, the number of technical replicates is given as 1 (which is a concern in

itself). While I do not doubt the trends observable in the data, the magnitudes might differ considerably. The authors already state this themselves, as the residence time difference shrinks quite a bit when changing experimental setup. This is not unusual for different measurement principles, but begs the question how big the difference between the ligands really are. Moreover, affinities are given as IC50s, which are inherently incomparable. Thus, the statement that both ligands bind with the same affinity is not entirely reproducible. Either the authors should convert to Ki values or at least ascertain that the compounds were measured on the same day in the same batch of experiments.

We thank the reviewer for pointing out the missing information of the number of replicates, which we have now included into the manuscript [“(ADP-Glo, ELISA, DLD1, HCT116, FP: n=3; SPR n=2; HotSpot, n=1)”]. Furthermore, now we provide the CI 95% is for the cell-based assays and SD for the enzymatic and FP assays in the updated figure (see below).

The comparability of the measured IC50 values is ensured by a reference compound (in our case SB203580) which is always part of the measurement and always within a certain range, otherwise the measurement is discarded. To calculate the Ki value based on IC50 is not useful since they were measured under the same conditions (same kinase, same ATP conc.) and had similar IC50 values so of course the calculated Ki is similar within one assay. Ki is supposed to compensate for differences between varying assay systems. However, this is not the case in tight binding situations in which the enzyme concentration significantly exceeds the inhibition constant. This thus leads to excessive Kis as can be seen with the ADP Glo assay. Ki's (\pm SD) was calculated based on the corrected cheng-prusoff equation: for ADP Glo: 6.7 ± 2.0 nM for **2** (IC50 = 26.7 ± 8.0 nM) and 7.8 ± 2.2 nM for **1** (IC50 = 31.3 ± 9.0 nM). ELISA Ki's $.0.6 \pm 0.2$ nM for **2** (IC50 = 1.8 ± 0.6 nM) and 1.1 ± 0.1 nM for **1** (IC50 = 3.4 ± 0.2 nM).

Figure 1. Compounds 1 and 2 biological activities and binding modes to p38 α MAPK. **a** 1 and 2 are structurally identical, except 2 has an additional thiophene-2-carboxamide group. **b** Biological activity of 1 and 2 in selected assays. Data for ELISA and SPR were reported earlier by our group^{27,28}. (ADP-Glo, ELISA, DLD1, HCT116, FP: n=3; SPR n=2; HotSpot, n=1). **c** Binding modes of 1 and 2 shown with representative snapshots obtained from MD simulations (the snapshots were selected to reflect the observed protein ligand interactions; see **d**). The key interaction residues are shown with sticks together with solvent molecules in the close proximity of the ligand. H-bonds shown with yellow, π -cation with red and π - π stacking with green dashed lines. **d** 2D-representation of protein–ligand interaction frequencies. Displayed are the most frequent H-bond, π -cation and π - π stacking interaction frequencies (interactions with > 15% frequency are shown).

* In my opinion, the authors should adapt the strength of their claims (“decisive role”, “extremely stable”, etc) to the strength of their arguments (see above). For the “extremely stable” statement, I’d like to see RMSD or RMSF plots, for instance, to judge for myself. We have now added these values in locations or revised the text to be more specific in a particular context, removing all potentially exaggerating adjectives. For instance, the new Supplementary Figure S3 displays the RMSD and RMSF values of **1** and **2**, from which can be observed that only the solvent exposed 2-morpholinoethylamide displays higher fluctuation in the simulations.

Supplementary Figure S3. Ligands 1 and 2 are stable in the simulations. (A) Root-mean-square fluctuation (RMSF) of **1** and **2** in the initial 10 μ s simulations. Only the solvent exposed 2-morpholinoethylamide displays high-fluctuation during the simulations. Similar RMSF was

observed also in the derived replica simulations (B) Root-mean-square deviation (RMSD) of **1** and **2** in the initial 10 μ s simulations. (C) RMSF of protein backbone in the initial 10 μ s simulations. Green vertical lines describe a ligand contact to the residue at some timepoint of the simulation. Secondary structure of the protein is described by blue and red shaded areas, which describe β -sheet and α -helix, respectively.

* The manuscript falls somewhat short of the claim made in the Introduction “We anticipated that a better understanding of this process can provide general principles, which could be applied in the compound design process when aiming for improved TRT.”. Assuming that the factors identified also hold true under the additional analyses mentioned above, what are tangible design recommendations to improve these particular ligands? Ideally, the authors would translate this into a novel ligand with ultra-long TRT themselves, but I realize that this is beyond the scope of this manuscript (but not beyond the abilities of the authors!).

We thank the reviewer for this certainly obvious argument. As candidate optimization for a clinical candidate, and this was the greater context of the program, is more a decathlon of e.g. activity, TRT, kinetics, tox-parameters, metabolism, physico-chemical properties etc., we followed a multi-parameter optimization rather than optimizing TRT alone. Within this context we do not see too much value for aiming solely for an ultra-high TRT. Nevertheless, we provide the suggestions for such an inhibitor by increasing the buried surface area, resolution barrier together with conformationally stabilizing the protein. These things can be applied in the CADD with MD simulations. Naturally, we do not know yet what is the minimum requirement for the timescale of the simulations that can provide useful and reliable results in this context. Therefore, this can be resolved by future case studies, and as the reviewer states that it is beyond this manuscript. We agree with the reviewer that an ultra-high TRT ligand would be scientifically interesting. This is indeed a stimulating question, if there is a limitation (a roof) in the TRT for a particular target (or even more specifically, a particular binding site). That is a bigger question, which certainly requires future research efforts and is beyond the scope of this manuscript.

* Personally, I find the structural figures quite crowded and hard to see what the authors intend to show me. Less information might be more here.

We appreciate the feedback and have now improved the clarity in all figures. Unfortunately, this is not an easy task as there is always a vast amount of data related to this type of long timescale simulations. Moreover, we do not prefer to cut the presented data extensively, as it would provide a false (or biased) view on the underlying processes.

Reviewer #2 (Remarks to the Author):

This manuscript by Pantzar and co-authors used mainly computational methods to study an interesting system. The two similar compounds, 1 and 2, bind to p38 α MAP kinase with the same K_d but very different kinetic behavior. The Laufer group has been studying this scaffold for a few years. Interestingly, this compound 2 has an additional function group that makes it become another type of inhibitors, and the group termed it type 1 1/2 inhibitor. While there are experimental structures and biochemical assays, the data cannot explain why the two have the same binding affinity but compound 2 has much slower k_{off} (also k_{on}). This study therefore carried out very long classical molecular dynamics simulations for the two complex systems and investigate protein dynamics and the role of water molecules. They also include some experimental results. The study about conformational differences of the MAPK when it binds to the two similar inhibitors is quite interesting. The analysis of water molecules is informative, too. The work utilized a commercial package, and the computation setup was reasonable. While the results are reliable, the current interpretation about conformational dynamics and long residence time (line 160) is not accurate and need to carefully revise.

It is very interesting that the results from SPR and FP are quite different. The measured residence time of Compound 2 (slower one) becomes faster when using FP, but the residence time of compound 1 (fast) becomes 10 times slower. Did the authors see similar trends for other inhibitors? Although it may be out of the scope of the paper, the authors may discuss more about it. Why does the presence of ATP slow down compound 1 unbinding so much? Is it possible that the discrepancy between SPR and FP is because of use of the flowing buffer in SPR?

We also found this difference interesting. We observed deviations of similar magnitude (in absolute terms) also for other compounds that are not reported in this manuscript. However, compounds systematically displaying either significantly higher or lower values in SPR respectively FP were not observed. Importantly, the trends remain intact between the two assays. In theory, a flowing buffer in SPR might decrease values due to dissociated compound being flushed away immediately, while in the FP assay rebinding might occur to a small degree. Also, immobilization of the kinase (in the SPR) can have an impact on compound binding in either way. Also, non-specific binding of compound might be more prominent in SPR (leading to higher values). As in FP assay p38 α MAPK is “similarly” in solution as in cells, this may be an important aspect why the biological results reflect more to this assay setting than to SPR. Interestingly, the ultra-fast residence time inhibitor SB203580 was (<LLOD) in our FP assay.

The simulations of only the bound states showed that the small chemical differences can lead to very different protein conformations. However, the reviewer cannot clearly see these differences from their figure 1 or other figures. Some figures that super-impose representative conformations or local energy minima from the two complexes should be helpful. This may also further explain why the water analysis using local energy minima showed different resolution energy.

We thank the reviewer for pointing out this unclarity. We have now included a new **Supplementary Figure S4** that should highlight these differences more clearly. Moreover, the actual MSM derived structures used for the calculations will be freely available via

Zenodo (together with all the simulations) to all interested readers who wish to pursue for a more detailed inspection.

Supplementary Figure S4. Comparison of the binding mode of 1 and 2. The shared scaffold appears in similar position, with only a small shift in the difluorophenyl orientation. The different DFG orientation reflects also to the activation loop (A-loop) configuration (open/closed), and thereby to the positions of these residues (e.g. Leu167). The shifted G-loop orientation in 2 shifts the locations of the residues in this region. Other regions and their residues in the binding site appear in comparable positions.

The entire simulations only sampled the inhibitor bound state, which are also pretty similar to their crystal structure bound conformations. Therefore, the major effort of this study is trying to get clues from the bound complex (see the copy/paste figure from the ref #8, lower right energy minima) to understand binding kinetics. It is not easy because the kinetic property is largely determined by the intermediate (transition) states. This simplified figure from their Ref #8 can be used to present their experiments that compound 2 (red solid line, longer residence time) should have larger energy barrier than that of compound 1 (black line).

[redacted]

Since the simulations are for the bound state, the conformational dynamics near the bottom of the energy well does not directly correlate to the energy barrier. Instead, based on thermodynamics, $G = H - ST$, if the bound state has more stable local energy minima, it may reduce entropic loss during binding. Even the constantly fluctuating conformations of the compound 1 complex might result in less favorable binding enthalpy, considering entropy, the system is not less stable. The data or thermodynamics/kinetics theories do not support that more equally stable local energy minima (actually they only showed 3 here which is very few) directly correlate to lower free energy barrier, or one dominant energy minima in the bound state means larger free energy barrier. The authors concluded that 3 more equally populated minima (compound 1, shorter residence time) vs 1 dominate minima (comp 2, longer residence time) in the bound state was too preliminary. Fig 2A and 3A cannot support their statements, and even the authors report the MSM kinetics info between those local energy minima is still useless. If they read their cited ref #41, that paper suggested that slower dissociation ligand has entropy driving binding and are more flexible, see their Abstract “Compounds bound to the helical conformation display slow association and dissociation rates, high-affinity and high cellular efficacy, and predominantly entropically driven binding. An important entropic contribution comes from the greater flexibility “. Therefore, while the resolution free energy approximation can be reasonably corrected to the free energy barrier, the very few local energy minima cannot bring any reasonable conclusion about the kinetic property.

We respectfully do not fully agree with the reviewers' statement that the bound conformation i.e. the bottom of the energy well does not directly correlate to the energy barrier. Although we agree with the reviewer that the transition states are important for the ligand dissociation energetics; however, this does not imply that the bound state is unrelated to these conformational states upon the ligand unbinding. First, the conformations in these intermediate transition states must be derived from the bound state. The protein cannot just arbitrarily jump from one conformation to another but must move via an energetically favorable route to the transition state, and therefore, the starting conformation (bound state) has a great impact on these transition state conformations. This bound state conformation may have no role in the ligand dissociation only case where a ligand is not enclosed to the protein (fully open binding site). Therefore, in the most cases (as in this context of the protein kinase) the protein conformation must have interplay with the ligand upon the dissociation process and for this reason the bound state has an impact on the transition state conformations and the energy barriers to reach to these transition conformations. For these reasons, we

cannot fully share the reviewer's view of that the bound state does not have any influence on the ligand dissociation. Furthermore, the previously introduced theory that the resolution barrier should affect ligand dissociation together with our findings with the resolution barrier energy differences do not agree with the reviewer's statement that the bound state should not have any effect on the ligand dissociation.

For the selection of only three states for the MSM, we would kindly ask to see our answer to the specific question of the states given by reviewer #3.

We have now included well-tempered metadynamics simulations to the study (a new section) that clearly demonstrate the higher energy barrier for ligand dissociation is associated with **cmpd 2** in these MSM derived conformations (Table 1, Supplementary Figures **S14–S31**: two examples of these Supplementary Figures also shown below).

Table 1. Observed ligand dissociation in well-tempered metadynamics simulations

Ligand dissociation ^a	1-S ₁	1-S ₂	1-S ₃	2-S ₁	2-S ₂	2-S ₃
Yes	59	54	53	8	5	9
No	1	6	7	52	55	51

^a Ligand was considered fully dissociated only if 15Å distance was reached during the simulation.

1

Supplementary Figure S22. Well-tempered metadynamics simulations of 1-S₃ conformation III (20 replicates). In the y-axis free energy and in the x-axis distance between the centre of mass of binding site residues and centre of mass of the inhibitor. Each simulation is 200 ns.

Supplementary Figure S26. Well-tempered metadynamics simulations of 2-S₂ conformation I (20 replicates). In the y-axis free energy and in the x-axis distance between the centre of mass of binding site residues and centre of mass of the inhibitor. Each simulation is 200 ns.

For the reviewer's statement related to the ref #41, we do not fully understand the reviewer's point. In our opinion, the ref #41 is not directly comparable here, as we do not observe any comparable conformational change in the p38 α MAPK as was presented in that paper i.e. helical – loop-like conformational changes. A recent paper (Berger et al. 2021), which also focusses on protein kinases is therefore in a better context, has similar conclusions related to the conformational stability with the ligand residence times. We have now cited this paper also in the discussion and adapted the text accordingly.

The updated part in the discussion is: *“It has been shown earlier that protein conformational flexibility plays a crucial role in the residence time⁴². In the context of protein kinases, it was recently discovered that inhibitors that supported a more ordered configuration of the protein were associated with a longer residence time⁴³.”*

Minor:

Page 4, line 76, the authors introduced the two inhibitors in this study. Their compound 1 is type I and compound 2 is type I/2 inhibitor, correct? The authors didn't mention it in the Abstract, and it may be better to at least mention it in Introduction.

We thank the reviewer for pointing out this and have now included this information in the introduction and into the abstract.

Updated introduction: "*Here we applied unbiased classical long timescale MD simulations to investigate behaviour of the protein–ligand complex of p38 α MAPK and two inhibitors, representing a first and a second generation dibenzosuberone-based inhibitor (type I and type I/2) originated from Skepinone-L²⁹*"

Reviewer #3 (Remarks to the Author):

The authors use molecular dynamics simulations to try to understand why the residence times of two structurally similar MAPK inhibitors differ dramatically. They provide a number of though provoking hypotheses that are of great importance and general interest. However, significant additional work is needed to make the arguments compelling.

My primary concern is whether the simulation results are anything more than correlative. Protein force fields have come a long way, but the reliability of the parameters one can find for arbitrary chemical compounds is questionable. Therefore, it is essential that the authors demonstrate that their simulations are consistent with experiments before trying to draw any conclusions from them. The most obvious candidate from my perspective is to measure the dissociation time computationally, and see if the rates are similar to those in experiment (at a minimum the simulations have to give the same rank order). The absolute timescales are far beyond reach of conventional MD simulations. However, there are a variety of methods that are likely up to the task and where code is freely available on github. I am specifically thinking of our own FAST adaptive sampling method, Alex Dickson's weighted ensembles approach, and Pratyush Tiwary's methods.

We thank the reviewer for pointing out this concern. To address this and to demonstrate that the force field performs well here in our setting, we have now included well-tempered metadynamics simulations to the study (see the new section: "*Well-tempered metadynamics simulations confirm higher dissociation barrier for long residence time inhibitor*"). We appreciate that the reviewer suggested methods here, but we still decided to conduct these simulations with Desmond that we would gain a direct answer for the reliability of the OPLS3e and our previous classical simulations. These simulations clearly demonstrate the

higher energy barrier for the dissociation with the longer residence time compound **2** (please see our answer to the reviewer #2 for some of the new data). Thus, we have a good agreement with the force field and the experiments. Furthermore, the used force field has been (and is) widely applied in the pharmaceutical industry; especially the force field has been enabling accurate results from the FEP+ calculations. Accuracy in these calculations is highly dependent on the force field and OPLS3 (and OPLS3e) have demonstrated their applicability. All in all, based on this we can rely on the force fields behavior in these simulations.

It is important to note that we were not aiming in this manuscript to predict the absolute dissociation time (as we try to imply in the abstract). In our opinion, we do not find it useful here, as it would be anyway conducted in a hindsight and is not a real prediction. In fact, we believe that it would be quite hard to make such prediction with the force field only, and not just tweaking and fitting the applied CVs to the data at hand. The reason for this is the following: for instance, as we have salt-bridges and cation- π interactions in this binding site; therefore, we anticipate that a polarizable force field would be more ideal (or a must) to enable such absolute predictions in this setting. However, as polarizable force fields are still not generally available (and validated to work in various settings, whereas the OPLS3e has been), we have not yet reached to the point of this type of absolute predictions. The important thing here is that our aim was/is not to predict the exact residence times but to find general concepts that are associated with a shorter/longer residence time, which can be potentially applied in future ligand discovery efforts. Nevertheless, we see that the new well-tempered metadynamics simulations included in the manuscript will provide more confidence on our findings and thank the reviewer for encouraging us to conduct these simulations.

My second major issue is the indirect nature of the comparison between the two complexes. Separate MSMs are built for the two different complexes, which makes direct comparison difficult. It would be interesting to build a common state space for the two datasets (e.g. define the states based on the union of the two datasets) and then estimate separate MSMs for each dataset so that the authors can directly assess which states are more/less populated in each case. This is an issue throughout the paper.

We totally agree with the reviewer that a direct comparison would be more straightforward approach. However, there occurs no interconversion between the datasets here as compound **2** is incompatible with the **1** associated p38 α MAPK conformation (steric constraints).

Therefore, they do not share such a common state space. We know from the experimental data that the binding of these compounds is different and anticipate that the inhibitor type may indeed be one of the key aspects defining the residence time of a compound. Obviously, these experimental structures are unable to capture the conformational dynamics and solvent-related aspects, which we aimed to cover in this manuscript. Nevertheless, we have now introduced an artificial comparison of the compounds by including new simulations of compound **1** in compound **2** associated p38 α MAPK conformation. Here we kindly ask the reviewer to refer to our answer to the reviewer #1, which discusses the key findings of these new simulations.

I also have a few minor issues that should be easier to address.

How were representative structures selected for each state? Given the authors coarse-grain extensively (to only 3 states), it seems like there is probably substantial conformational heterogeneity within each state. Building higher resolutions MSMs with more states would be informative.

We totally agree with the reviewer with the point that there is conformational heterogeneity within each of the three states. The representative structures for the states were derived using the default approach in PyEMMA (pcca_samples). As the choice of the number of states is partially based on the user (must pass the validation), and a higher resolution MSM could be perhaps possible. However, our focus in this manuscript is on a ligand discovery setting and we try to find out more generalizable aspects that could be potentially translated to other settings and not only to this case. We believe that this aim would be more easily accomplished with fewer states. In the case of e.g. 10 states (with a higher resolution MSM) there would be an abundance of data and it would be less easy to derive a more generalizable results, and one would be easily lost in the subtle details of each state. Moreover, the other aspect is the practical perspective. If the key findings can be easily derived from just a few states, this approach can be more effortlessly be applied for a different set of simulations. Therefore, we do not really see the usefulness for building higher resolution MSM to obtain more specific results for this specific case, meanwhile most probably losing the more generalizable aspects. Overall, as our main aim in the long-term is to utilize this type of approach in the compound design, we are more interested in a more "macro state" kind behavior, for which the fewer states is a better approach.

I also strongly suggest that the authors minimize the use of unnecessary acronyms, like TRT and cmpds.

These have been now changed according to the reviewer's suggestion throughout the manuscript.

It is unclear how the structures shown in Figure 1 were selected. No statistical evidence is given that they are representative.

Those structures were selected based on the observed interaction frequencies (shown in panel D) and based on the interaction frequencies these structures should be representative. Their aim here is in this provide a 3D-aspect for the 2D-interactions and vice versa. We have now included the clarification into the figure legend: “(the snapshots were selected to reflect the observed protein ligand interactions; see D)”.

Finally, I would like to (partly selfishly) suggest the following review article be cited for the use of MSMs to address problems like this one:

Advanced Methods for Accessing Protein Shape-Shifting Present New Therapeutic Opportunities

We have included this in the manuscript to the beginning of the MSM-related section: “This methodology is able to capture relevant long timescale kinetic conformational states, the metastable states, of the protein–inhibitor complex^{32–34}.”

Greg Bowman, Washington University

REVIEWERS' COMMENTS

Reviewer #1 (Remarks to the Author):

My concerns have been answered by the authors in convincing ways. A small criticism is still that an $n=1$ for the HotSpot measurement is too little, but as the precise numbers are not relevant to the message of the manuscript and the results qualitatively agree with the other experiments, I think that this can be accepted, as it is also clearly spelled out now in the text.

Reviewer #3 (Remarks to the Author):

The authors addressed my biggest concern